# SHADOWSPEAK: IS IT POSSIBLE TO COMMUNICATE CROSS-ROOM SOLELY BY DECODING GESTURE SHADOWS?

## ABSTRACT

Accurately decoding hidden information in dynamic shadows for Non-Line-of-Sight (NLOS) imaging enables us to overcome visual occlusions and perceive or reconstruct obscured targets. This breakthrough holds significant potential for real-world applications such as disaster rescue, autonomous driving, and security surveillance. Conventional algorithms struggle to model the physical propagation of light in space. Furthermore, the signal distortions introduced by nonlinear transformations incur the loss of geometric information about the source scene, limiting sensitivity to subtle shadow variations. To overcome these challenges, we present Radiation-constraint Network (RacoNet) that marries physical propagation simulation with geometric-information recovery to interpret minute gesture signals embedded in dynamic shadows. In RacoNet, Radiance-Constrained Light-Transportation (RCLT) optical propagation is proposed to capture complete light-space information. Meanwhile, Geometric Information Aliment Operation (GIAO) restores source-scene geometry lost in the modulated shadow through layer-by-layer refined prior attention. Moreover, Kolmogorov-Arnold Enhanced Layerwise Nonlinear Reorganization (KA-ELNR) fuses light-space and geometric cues to produce the final decoded output. Extensive experiments show that RacoNet markedly surpasses existing approaches in both accuracy and robustness for dynamic-shadow decoding, confirming the possibility of gesture-based information interaction via shadows.

## 1 INTRODUCTION

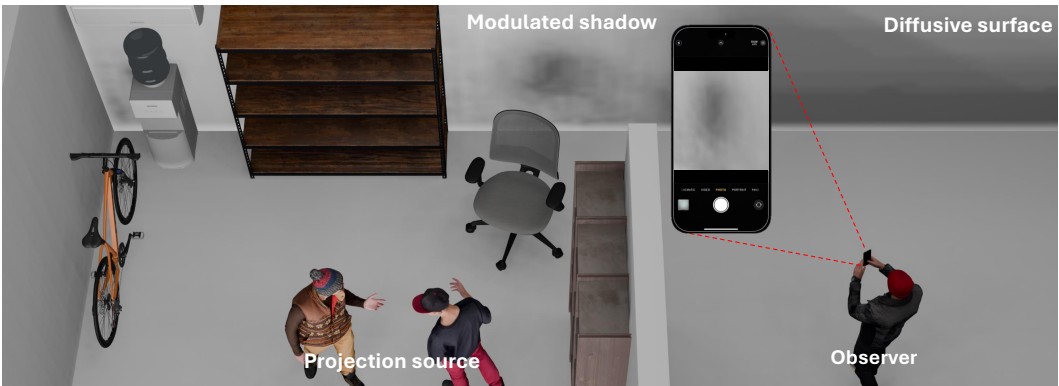

Figure 1: Scenario diagram. *Projection source* encodes gesture semantics into a spatiotemporally *Modulated shadow* projected onto a passive *Diffusive surface*. *Observer* recovers the original semantic information by observing the Modulated shadow.

In the evolving domain of Non-Line-of-Sight (NLOS) imaging and covert communicationAsadi-Aghbolaghi et al. (2017), the ability to decode obscured information from indirect optical signatures presents a critical challenge with profound implications for secure data transmission. This work posits a novel inquiry: Can subtle variations in dynamic shadows serve as a robust medium for occlusion-tolerant, tamper-resistant communication? Traditional NLOS methods based on direct

optical sensing or acoustic channels remain vulnerable to interception and environmental interferenceWang et al. (2021). Here, we explore a fundamentally distinct paradigm: encoding gestural semantics into spatially consistent modulated shadow patterns projected onto passive diffusive surfaces (Figure 1). An observer, situated in a non-adjoining space, can theoretically decipher these latent signatures by analyzing radiometric fluctuations to effectively transform ambient shadows into an information-theoretically secure communication channel. This approach capitalizes on the spatial coherence between shadow-casting source and observation plane, bypassing traditional electromagnetic or acoustic pathways that are susceptible to eavesdropping.

Although this scheme could be highly confidential, accurately interpreting the modulated shadow is still a difficult issue. The main reasons lie in the following three points: (i) the light and shadow on the wall form a superposed state of the light fields from the target and from extraneous objects, and diffuse-reflection transport is anisotropic, so existing methods struggle to accurately model its propagation and to demodulate the target signal; (ii) the camera exhibits a nonlinear response during capture, namely pixel values are not proportional to the irradiance at the entrance pupil, which amplifies inversion errors and uncertainties and in turn causes the modulated shadow to lack recoverable source-space geometry; (iii) algorithms and traditional models find it difficult to effectively combine the spatial physical propagation of light with the source-space geometry.

In response to the above problems, this paper proposes a Radiation-constraint Network (RacoNet). In our research, we found that the spatial physical propagation of light can be decomposed into axial transmission and diffuse transmission. Therefore, we constructed a two-stream *Radiance-Constrained Light-Transportation (RCLT)*, established a high-order optical joint encoding between axial transmission and diffuse transmission, simulated the spatial physical propagation of light, and thus extracted the light space information of the modulated shadow. Secondly, for the problem of missing source space geometry in the modulated shadow, we introduced *Geometric Information Complementation Operation (GIAO)* to gradually refine the missing latent space prior in the modulated shadow distribution, thereby recovering the source space geometry information. Finally, in order to combine the extracted light space information with the source space geometry information, we constructed *Kolmogorov-Arnold Enhanced Layerwise Nonlinear Reorganization (KA-ELNR)*. It operates on the fused feature domain through subspace decomposition nonlinear blocks and hierarchical combination mapping, so that the local spectral transformation can be cumulatively aligned to the semantically separable output manifold. To verify the feasibility of the above scheme, we evaluated the RacoNet on three datasets and achieved relatively ideal results. The main contributions of this paper are summarized as follows:

- We established RacoNet to simulate light transport and decrypt modulated shadows. Through extensive qualitative and quantitative experiments on the above dataset, the results show that RacoNet outperforms other state-of-the-art models in accurately decoding hidden information within dynamic shadows, enabling effective cross-room communication solely via decoded shadow of gesture.

- We proposed RCLT, in which dual-path Transformer branches, hierarchically stratified by frequency-domain modulation and multiscale attention, formulate orthogonal embeddings for axial and scattered photonic trajectories; this mechanism compensates for conventional insufficiencies in modeling transport across axial transmission and scattering transmission.

- We proposed GIAO to deploy depth-aware local-perceptual stratification alongside lightweight multi-head attention filters. It recovers spatial priors lost in modulated shadow measurements by reconstituting latent geometrical structure of the occluded projection source, thereby circumventing limitations induced by source-domain topological underrepresentation.

- We proposed KA-ELNR framework to utilize localized nonlinear activations, derived through a blockwise decomposition process and recursive fusion hierarchies. This mechanism enables subspace-aligned semantic refinement, facilitating cumulative abstraction. The approach integrates light space information with the geometry of the source space, leveraging a theorem-driven process that incorporates the recursive structure for efficient fusion and refinement of high-dimensional data.

## 2 RELATED WORKS

### 2.1 NON-LINE-OF-SIGHT IMAGING

NLOS imaging infers occluded geometry or semantics by analyzing indirect radiative fields. Approaches are classified by sensor type:

**Long-Wave Infrared NLOS Imaging.** Thermal emission in the 8–14 μm band serves as active illumination Maeda et al. (2019b); Jin et al. (2025). Liu et al.Liu et al. (2023) combine LWIR intensity and polarization gradients in a bifurcated deep network for precise reconstructions. Maeda et al.Maeda et al. (2019a) propose a first-order scattering transport model with emissive priors to constrain inversion and stabilize output.

**Photon-Counting NLOS Imaging.** Time-correlated single-photon counting (TCSPC) achieves picosecond resolution in photon-starved settings Li et al. (2021); Czerwinski (2022). Wang et al.Wang et al. (2024) model transients as Poisson convolutions with known IRFs and invert them iteratively. Sultan and DoveSultan et al. (2024) unify ToF and occluder-shadow cues via a dual-domain Wigner framework. Ding et al.Ding et al. (2024) enforce curvature regularization in object and transform spaces, while Li et al.Li et al. (2022b) exploit first-photon statistics for robust real-time inference.

**Camera-Driven NLOS Imaging.** Conventional cameras enable low-cost NLOS. Liu et al.Liu et al. (2024a) use chromatic differential correlation with low-coherence speckles for single-shot capture. Zhu et al.Zhu et al. (2024) leverage event cameras' temporal sparsity to track hidden dynamics. Czajkowski and Murray-BruceCzajkowski & Murray-Bruce (2024) integrate spectral capture and implicit scene priors to reconstruct 3D volumes from diffuse relay surfaces.

### 2.2 GESTURE RECOGNITION

Gestures are intentional, structured body movements, usually of the hands and arms, that convey meaning or commands Studdert-Kennedy (1994); Kendon (2004); Dukauskaite (2024); Kandana Arachchige et al. (2021). Gesture recognition is the computational process of detecting, tracking, and interpreting these movements to infer intent or communication. It enables natural touchless interaction in applications such as sign-language translation Núñez-Marcos et al. (2023), virtual and augmented reality Gavgiotaki et al. (2023); Liu et al. (2024b), robotics Chen et al. (2024), and ambient intelligent environments Dunne et al. (2021). Methods are broadly classified as computer vision–based Tripathi & Verma (2024); Aggarwal et al. (2023); Cao et al. (2022) or sensor-based Chen et al. (2022); Tchantchane et al. (2023); Sosin et al. (2018). Challenges such as occlusion, user variability, and robustness to illumination and background continue to drive research on adaptable recognition frameworks.

## 3 METHODOLOGY

NLOS gesture recognition, predicated on the photometric interpretation of modulated light fields projected onto diffuse relay surfaces, conventionally resorts to deep convolutional methods for direct label regression from such projections. These methods, however, treat the projections as static radiometric textures devoid of underlying transport context. Therefore, they fail to encode either the intrinsic geometry of the occluded projection source or the governing photonic propagation physics so linear axial transmission and higher-order indirect scattering remain unmodeled. The absence of physically informed constraints within these mappings precludes integration of progressive semantic hierarchies across layers, hindering discriminative inference. The overall architecture diagram is shown in the Figure 2.

### 3.1 RADIANCE-CONSTRAINED LIGHT-TRANSPORTATION

The photonic modulation observed in NLOS imaging emerges as an entangled consequence of dual-modal optical interactions—namely, direct axial linear radiative transfer and higher-order scattering-induced nonlinear deviations—each contributing distinct yet co-implicated propagation signatures. Current convolutional architectures, limited by the inherently localized receptive aggregation and spatially invariant kernel design, fail to encode the volumetric diffusion and nonlocal photometric dependencies intrinsic to such propagation manifolds. Moreover, pooling and strided reduction aggravate representational degradation, thereby diminishing the semantic separability of modulated shadows. Absent global spatial coherence and frequency-sensitive modeling, these networks yield reconstructions that elide the underlying physics of light transport.

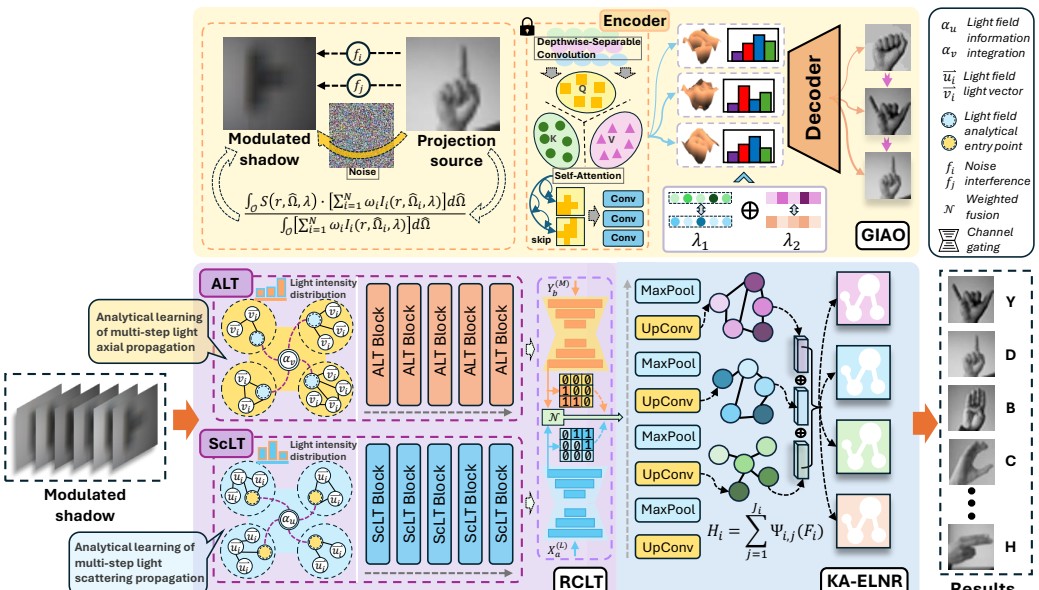

Figure 2: The overall architecture of RacoNet. The modulated shadow (generated by projecting gestures through multipath radiation transmission) is provided to RCLT. In the generation process of this shadow, each reflection or scattering of light will produce a new light field. Encoder comprehensively analyzes the axial transmission and scattered-transmission light fields through a multi-branch multi-layered structure to restore the physical process of light propagation. GIAO then uses hierarchical depth-aware filtering to restore the occluded source geometry clues. KA-ELNR performs subspace-specific nonlinear fusion of radiation information and geometric structure information. Finally, RacoNet outputs synthesis of potential gesture information under NLOS conditions.

Conversely, Transformer-based formulations, owing to attention-induced dynamic connectivity, exhibit marked efficacy in modeling cross-spatial, scale-agnostic light flow dependencies. To exploit this potential, a bifurcated architecture is formulated (Figure 2), which contains two structurally disjoint yet semantically coupled branches: Scattering Light-Transportation (ScLT) and Axial Light-Transportation (ALT)—respectively instantiating nonlinear volumetric interactions and linear radiative trajectories. This separation enables independent encoding of scattering and axial modalities, while their integration facilitates cross-modal fusion within a unified latent space. The architectural duality enforces orthogonal inductive biases: frequency-domain spectral reconstitution and reparameterized attention in ScLT, versus scale-aware axial attention pyramids in ALT.

Finally, cross-branch feature alignment, enforced via resolution-preserving interpolation or stratified coupling, culminates in a joint representation space that maintains fidelity to both local photonic distortions and globally consistent radiometric transport—thereby forming a frequency-adaptive, propagation-constrained embedding conducive to subsequent geometric-physical decoding.

### 3.1.1 SCATTERING LIGHT-TRANSPORTATION

The ScLT branch, transformed under a Transformer-derived framework, incorporates layerwise reparameterized kernels and spectral-domain modulation, with the expectation of approximating the nonlinearities induced by multi-bounce reflection, volumetric scattering, and diffractive perturbation.

Suppose $\Theta^{(0)} \in \mathbb{R}^{\alpha \times \beta}$ denote the input tensor, where $\alpha$ and $\beta$ represent its initial height and width, respectively. We first expand the input tensor via an outer product decomposition of its row and column components. This outer product expansion is given by

$$\Theta^{(0)} = \underbrace{\left[\Theta_{1,1}^{(0)} \Theta_{1,2}^{(0)} \cdots \Theta_{1,\beta}^{(0)}\right]^T}_{\text{Row 1}} \otimes \underbrace{\left[\Theta_{2,1}^{(0)} \Theta_{2,2}^{(0)} \cdots \Theta_{2,\beta}^{(0)}\right]^T}_{\text{Row 2}} \otimes \cdots \otimes \underbrace{\left[\Theta_{\alpha,1}^{(0)} \Theta_{\alpha,2}^{(0)} \cdots \Theta_{\alpha,\beta}^{(0)}\right]^T}_{\text{Row } \alpha},$$

(1)

where each element $\Theta_{i,j}^{(0)}$ corresponds to the pixel or feature value at row $i$ and column $j$ of the original input.

Figure 3: ScLT Block first applies multi-angle directional attention to the input feature map for emulating the nonlinear volumetric scattering of multi-bounce light, and then uses downsampling, pooling, and FFT-based spectral modulation to extract frequency-domain propagation cues. A hierarchical feature reconfiguration module subsequently integrates these spectral representations via learnable convolutions, restoring both local photonic distortions and global radiometric coherence.

In each layer $m \in \{1, \ldots, M\}$, we apply the directional attention operator $\mathcal{O}_{\|}(\cdot)$ to the previous layer's output $\Theta^{(m-1)}$ along the $\xi$ or $\eta$ direction to capture local dependencies. Specifically, let $\{f_k\}_{k=1}^K$ be a set of directional filters corresponding to angles $k$. For each spatial position $\mathbf{x}$, we compute the filter responses

$$r_k(\mathbf{x}) = \left(f_k * \Theta^{(m-1)}\right)(\mathbf{x}), \tag{2}$$

and normalize via softmax to obtain attention weights

$$\alpha_k(\mathbf{x}) = \frac{\exp\left(r_k(\mathbf{x})\right)}{\sum_{j=1}^K \exp\left(r_j(\mathbf{x})\right)}. \tag{3}$$

The operator $\mathcal{O}_{\|}$ then aggregates these weighted responses across all directions using the tensor product $\otimes$, yielding the attention mapping:

$$\Delta^{(m)} = \mathcal{O}_{\|}\left(\Theta^{(m-1)}\right) = \begin{bmatrix} \Delta_{1,1}^{(m)} \\ \Delta_{1,2}^{(m)} \\ \vdots \\ \Delta_{1,\beta_m}^{(m)} \end{bmatrix} \otimes \begin{bmatrix} \Delta_{2,1}^{(m)} \\ \Delta_{2,2}^{(m)} \\ \vdots \\ \Delta_{2,\beta_m}^{(m)} \end{bmatrix} \otimes \cdots \otimes \begin{bmatrix} \Delta_{\alpha_m,1}^{(m)} \\ \Delta_{\alpha_m,2}^{(m)} \\ \cdots \\ \Delta_{\alpha_m,\beta_m}^{(m)} \end{bmatrix}. \tag{4}$$

Next, the downsampling operator $\mathcal{D}(\cdot)$ and the intra-layer pooling operator $\mathcal{P}(\cdot; \omega_s^{(m)})$ are jointly applied to $\Delta^{(m)}$ to produce multi-scale representations

$$\Theta_s^{(m)} = \mathcal{P}\left(\mathcal{D}(\Delta^{(m)}); \omega_s^{(m)}\right), \quad s \in \{1, \ldots, S_m\}, \tag{5}$$

where the parameter $\omega_s^{(m)} \in \mathbb{R}^k$ governs the pooling characteristics at scale $s$, ensuring that each $\Theta_s^{(m)} \in \mathbb{R}^{\alpha_m^{(s)} \times \beta_m^{(s)}}$ reflects the resolution decomposition.

To incorporate frequency-domain features, a Fourier mapping with spectral modulation $\mathcal{F}(\cdot; \zeta^{(m)})$ is executed on each scale representation, yielding ($\Xi^{(m,s)} \in \mathbb{R}^{\alpha_m^{(s)} \times \beta_m^{(s)}}$)

$$\Xi^{(m,s)} = \mathcal{F}\left(\Theta_s^{(m)}; \zeta^{(m)}\right) = \begin{bmatrix} \Xi_{1,1}^{(m,s)} \\ \Xi_{1,2}^{(m,s)} \\ \vdots \\ \Xi_{1,\beta_m^{(s)}}^{(m,s)} \end{bmatrix} \otimes \begin{bmatrix} \Xi_{2,1}^{(m,s)} \\ \Xi_{2,2}^{(m,s)} \\ \vdots \\ \Xi_{2,\beta_m^{(s)}}^{(m,s)} \end{bmatrix} \otimes \cdots \otimes \begin{bmatrix} \Xi_{\alpha_m^{(s)},1}^{(m,s)} \\ \Xi_{\alpha_m^{(s)},2}^{(m,s)} \\ \vdots \\ \Xi_{\alpha_m^{(s)},\beta_m^{(s)}}^{(m,s)} \end{bmatrix}, \tag{6}$$

where $\zeta^{(m)} \in \mathbb{C}^\ell$ characterizes the frequency modulation (More details in Appendix A.6).

Then, within a local region $\Omega^{(s)} \subset \mathbb{R}^2$, an adaptive kernel function $\kappa^{(m)}(\xi, \eta; \lambda_s^{(m)})$, with parameters $\lambda_s^{(m)} \in \mathbb{R}^d$ that adaptively adjust its shape, is employed to perform double integration on the

frequency features. This produces the locally weighted aggregation result

$$\Psi^{(m,s)} = \iint_{\Omega^{(s)}} \kappa^{(m)}(\xi,\eta;\lambda_s^{(m)})\,\Xi^{(m,s)}(\xi,\eta)\,d\xi\,d\eta, \tag{7}$$

where $\Psi^{(m,s)} \in \mathbb{R}^{\gamma_m}$ reflects the aggregated local features with weighting. Subsequently, the multi-scale outputs are fused by weighting and summing across scales using scalar weights $\omega_s^{(m)} \in \mathbb{R}$ (satisfying $\sum_{s=1}^{S_m} \omega_s^{(m)} = 1$) to form the global representation at layer $m$, is given by:

$$\widehat{\Theta}^{(m)} = \sum_{s=1}^{S_m} \omega_s^{(m)}\,\Psi^{(m,s)} = \sum_{s=1}^{S_m} \omega_s^{(m)}\left[\sum_{(\xi,\eta)\in\Omega^{(s)}} \kappa^{(m)}(\xi,\eta;\lambda_s^{(m)})\,\Xi^{(m,s)}(\xi,\eta)\right]. \tag{8}$$

Finally, the fusion representations from all layers $\{\widehat{\Theta}^{(m)}\}_{m=1}^{M}$ are input into a cross-layer fusion mapping $\mathcal{T}(\cdot;\eta)$ to obtain the ultimate global output embedding

$$\Upsilon = \mathcal{T}\Big(\{\widehat{\Theta}^{(m)}\}_{m=1}^{M};\eta\Big), \tag{9}$$

where the parameter $\eta \in \mathbb{R}^p$ controls the fusion strategy, ensuring that $\Upsilon \in \mathbb{R}^{\delta}$ encapsulates both spatial and frequency information. The entire branch—starting from the outer product expansion of the initial input, proceeding through the directional attention mapping, multi-scale decomposition and pooling, Fourier-domain modulation, local adaptive integration, and multi-scale weighted fusion, and culminating in cross-layer aggregation—effectively captures both direct and indirect propagation characteristics while enabling efficient computation.

### 3.1.2 AXIAL LIGHT-TRANSPORTATION

The ALT stream, structured upon Transformer-based architecture, prioritizes directional attention mechanisms to explicitly model spatially anisotropic and directionally persistent radiative interactions. Let the initial input feature be expanded as $\mathbf{Y}^{(0)} = \left(y_{i,j}^{(0)}\right)_{j=1\cdots W'}^{i=1\cdots H'} \in \mathbb{R}^{H'\times W'}$, where $y_{i,j}^{(0)}$ denotes the initial input signal at the spatial location $(i,j)$.

For the $m$th layer ($m = 1,\ldots,M$), first introduce the axis-parallel attention operator $\mathcal{O}_\beta(\cdot)$ with a directional parameter $\beta$, which is applied to the previous layer's output to obtain

$$\widetilde{\mathbf{Y}}^{(m)} = \mathcal{O}_\beta\big(\mathbf{Y}^{(m-1)}\big) = \begin{bmatrix} \alpha_{1,1}^{(m)} & \alpha_{1,2}^{(m)} & \cdots & \alpha_{1,J_m}^{(m)} \\ \beta_{2,1}^{(m)} & \beta_{2,2}^{(m)} & \cdots & \beta_{2,J_m}^{(m)} \\ \vdots & \vdots & \ddots & \vdots \\ \gamma_{I_m,1}^{(m)} & \gamma_{I_m,2}^{(m)} & \cdots & \gamma_{I_m,J_m}^{(m)} \end{bmatrix}, \tag{10}$$

where each element $\alpha_{i,j}^{(m)}$, $\beta_{i,j}^{(m)}$, $\gamma_{i,j}^{(m)}$ represents a feature component obtained after different directional modulations, and $I_m$ and $J_m$ denote the row and column dimensions after the attention modulation.

Then, to encode scale-diverse axial transport information, resolution-decomposed representation, introducing the downsampling matrix $\mathbf{D}^{(m)}$ and, for each scale index $s$ ($s = 1,\ldots,S_m$), a pooling matrix $\mathbf{\Pi}^{(m,s)}$. By applying the matrix transformation to $\widetilde{\mathbf{Y}}^{(m)}$, we obtain

$$\mathbf{Y}^{(m,s)} = \mathbf{\Pi}^{(m,s)}\Big(\mathbf{D}^{(m)}\cdot\widetilde{\mathbf{Y}}^{(m)}\Big) = \begin{bmatrix} \phi_{1,1}^{(m,s)} & \phi_{1,2}^{(m,s)} & \cdots & \phi_{1,L_{m,s}}^{(m,s)} \\ \psi_{2,1}^{(m,s)} & \psi_{2,2}^{(m,s)} & \cdots & \psi_{2,L_{m,s}}^{(m,s)} \\ \vdots & \vdots & \ddots & \vdots \\ \omega_{K_{m,s},1}^{(m,s)} & \omega_{K_{m,s},2}^{(m,s)} & \cdots & \omega_{K_{m,s},L_{m,s}}^{(m,s)} \end{bmatrix}, \tag{11}$$

where $\phi_{i,j}^{(m,s)}$, $\psi_{i,j}^{(m,s)}$, and $\omega_{i,j}^{(m,s)}$ are the feature coefficients sampled locally at scale $s$, and $K_{m,s}$ and $L_{m,s}$ are the corresponding matrix dimensions.

Within each scale $s$, consider a local receptive field $\Lambda^{(m,s)}$ over which spatial aggregation is performed via an attention kernel matrix. Define $\mathbf{A}^{(m)}(\tau,\omega) = \left[\xi_{\mu,\nu}^{(m)}(\tau,\omega)\right] \in \mathbb{R}^{P\times Q}$, where $\xi_{\mu,\nu}^{(m)}(\tau,\omega)$ denotes the modulation coefficient at the spatial location $(\tau,\omega)$, with the indices $\mu,\nu$ running from 1 to $P$ and 1 to $Q$, respectively. Aggregating over the region $\Lambda^{(m,s)}$ using a discrete

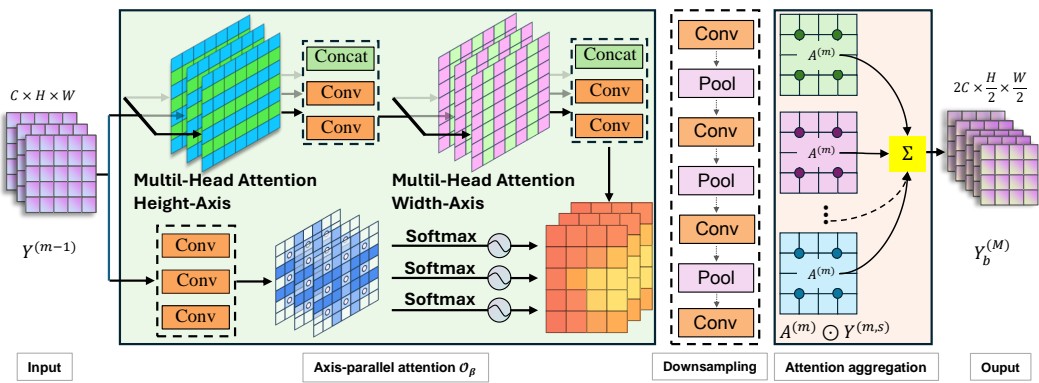

Figure 4: ALT Block architecture, in which separate height-axis and width-axis multi-head attention streams capture anisotropic axial light transport by encoding directional radiative cues along vertical and horizontal planes. Each stream incorporates downsampling and pooling operators to simulate hierarchical discretization of radiative kernels, aggregating illumination features at varying resolutions to approximate physical light propagation. Then, an attention-aggregation fuses these directional feature maps into a unified radiance-conditioned embedding.

integral gives

$$\mathbf{Z}^{(m,s)} = \iint_{\Lambda^{(m,s)}} \mathbf{A}^{(m)}(\tau,\omega) \odot \mathbf{Y}^{(m,s)}(\tau,\omega) \, d\tau \, d\omega, \tag{12}$$

which, after discretization, can be written as

$$\mathbf{Z}^{(m,s)} = \begin{bmatrix} \sum_{(\tau,\omega)} \xi_{1,1}^{(m)}(\tau,\omega) \, \phi_{1,1}^{(m,s)}(\tau,\omega) & \cdots & \sum_{(\tau,\omega)} \xi_{1,L_{m,s}}^{(m)}(\tau,\omega) \, \phi_{1,L_{m,s}}^{(m,s)}(\tau,\omega) \\ \vdots & \ddots & \vdots \\ \sum_{(\tau,\omega)} \xi_{P,1}^{(m)}(\tau,\omega) \, \omega_{K_{m,s},1}^{(m,s)}(\tau,\omega) & \cdots & \sum_{(\tau,\omega)} \xi_{P,L_{m,s}}^{(m)}(\tau,\omega) \, \omega_{K_{m,s},L_{m,s}}^{(m,s)}(\tau,\omega) \end{bmatrix}, \tag{13}$$

where $\odot$ denotes element-wise multiplication and $(\tau,\omega) \in \Lambda^{(m,s)}$.

Then, fusion across channels is performed by summing the aggregated matrices from all scales: $\widehat{\mathbf{Y}}^{(m)} = \sum_{s=1}^{S_m} \mathbf{Z}^{(m,s)}$.

After recursively applying these operations over $M$ layers, the final axial output matrix is given by $\mathbf{Y}_b^{(M)} = \widehat{\mathbf{Y}}^{(M)}$, which structurally encapsulates a scale-sensitive discretization of linear radiative transfer, analogous to the multi-level finite approximations of Helmholtz-type dynamics (Proof in Appendix A.2).

Finally, to achieve aligned fusion with the feature matrix $\mathbf{X}_a^{(L)} \in \mathbb{R}^{U \times V}$ obtained from another branch (the scattering module), an interpolation-matching based terminal fusion operator $\mathcal{M}(\cdot, \cdot)$ is employed, expressed as

$$\mathbf{H} = \mathcal{M}\left(\mathbf{X}_a^{(L)}, \mathbf{Y}_b^{(M)}\right) = \left[\mathbf{X}_a^{(L)} \quad \| \quad \mathbf{Y}_b^{(M)}\right], \tag{14}$$

where $\|$ denotes either column-wise concatenation or pointwise integration.

## 3.2 GEOMETRIC INFORMATION ALIMENT OPERATION

Latent geometric visual attributes, referring exclusively to object shape, relative spatial disposition, and inter-object topological configuration, are herein distinguished from semantic abstraction, being purely structural in essence. In the context of NLOS imaging, photonic propagation is invariably subjected to multifold perturbations such as reflection, refraction, volumetric scattering, and diffractive interference. Each of them collectively induces stochastic deformations in both amplitude and phase of incident light. Therefore, disparate scene geometries often give rise to near-indistinguishable modulated shadows, thus rendering the inverse mapping problem markedly ill-posed. We use hierarchical encoding to recover the suppressed geometric cues that are missing in the projection manifold, with the expectation of addressing this degeneracy.

This approach employs a multi-stage hierarchical encoding to recover structural cues lost in indirect projections. Given an input tensor $X \in \mathbb{R}^{B \times C_{\text{in}} \times H \times W}$, we first apply a series of convolutions, normalizations, and activations to obtain initial features. Let $\mathcal{F}_3(\cdot)$ denote a $3 \times 3$ convolution operator, $\mathcal{N}(\cdot)$ denote batch normalization, and $\sigma(\cdot)$ denote the GELU activation. To simplify the notation, define $\mathcal{S}(Z) = \sigma\big(\mathcal{N}(\mathcal{F}_3(Z))\big)$.

Then, the low-level feature map $X'$ is given by $X' = \mathcal{S}^{\circ 3}(X)$. This step maintains spatial resolution while enriching local context.

Subsequently, each stage $i$ takes $X_i$ as input and first performs depthwise-separable convolutions to aggregate local context, yielding $X_i' = X_i + (K * X_i)$, where $*$ denotes depthwise-separable convolution with a learnable kernel $K$. Next, a lightweight multi-head self-attention module (enabled only at lower resolutions) captures nonlocal dependencies, followed by a channel-wise feedforward integration in a reverse residual manner. Denoting the output after these operations as $X_i'''$, a strided convolution-based projection then reduces spatial resolution and increases channel capacity: $X_{i+1} = \mathcal{P}(X_i''')$. Iterating across $N$ stages, we ultimately obtain $X_{\text{out}} = \mathcal{P}(X_N)$.

The inclusion of both local and global interactions at progressively reduced resolutions reinforces geometric factors otherwise missing from the raw projection manifold. Implementation details are provided in the Appendix A.4.

### 3.3 Kolmogorov-Arnold Enhanced Layerwise Nonlinear Reorganization

We introduce a hierarchical model that partitions the input space into localized submanifolds, applies nonlinear transformations within each partition, and recombines these intermediate representations into a global latent descriptor. Building on Kolmogorov–Arnold theory, we replace the usual monolithic MLP structure with independent, locally adaptive transformations that yield semantically responsive features. First, we decompose the input into multiple disjoint subspaces and apply separate nonlinear mappings to each. We then fuse the resulting partition-wise descriptors into higher-order embeddings and concatenate these embeddings to form a unified representation. In addition, our framework extends beyond standard linear projections by incorporating local spline expansions to adaptively modulate each channel. These spline-based transformations are stacked across multiple layers using residual shortcuts, which is expected to enable the network to capture fine-grained local nonlinearities at various depths. Finally, we compress the resulting high-dimensional representation into the desired output dimension, preserving discriminative flexibility. Implementation details, including formal definitions of the partitioning operator, integrals over submanifolds, spline parameterizations, and weight matrices, are provided in the Appendix A.5.

## 4 Experiments and Analysis

### 4.1 Datasets

Three datasets were constructed on three public available datasets (more details@Appendix A.3):

**Sign Language for Numbers (S-Numbers)**: Sign Language for Numbers, synthesized through a photometric simulation framework, parameterized by radiative transfer approximations modeling multipath photon propagation under NLOS constraints. Modulated shadows, derived by forward-solving the Radiative Transfer Equation (RTE) for varying gestural inputs (Appendix A.1), formed the sample basis.

**Sign Language MNIST (S-MNIST)**: based on Sign Language MNIST. The process of generating modulated shadows is the same as S-Numbers.

**Sign Language MNIST measured (S-MNISTm)**: based on Sign Language MNIST (same as the above), empirical measurements collected from a controlled testbed, and a DFK-33UX183 industrial camera as receiver. The projected gesture-bearing signals underwent diffuse reflection off an intermediary wall surface; the modulated light distribution captured downstream constituted the empirical corpus of the dataset.

### 4.2 Performance of RacoNet

To assess the functional viability of the RacoNet, comparative evaluations were performed against multiple deep neural architectures across three gesture recognition datasets, each constructed under NLOS constraints reported in Table 1. RacoNet consistently outperformed baselines, not due

to isolated architectural components, but owing to its compound encoding strategy. RacoNet's Radiance-Constrained Light-Transportation explicitly disentangles linear (axial) propagation from higher-order volumetric scattering to capture the true photonic transport dynamics that conventional convolutions alias into noise. The Geometric Information Aliment Operation then hierarchically restores the source-scene geometry lost to nonlinear shadow modulation, reintroducing the spatial priors suppressed by diffuse reflections. Finally, the Kolmogorov–Arnold Enhanced Layerwise Nonlinear Reorganization fuses these radiometric and geometric cues across scales, preserving both structural coherence and spectral fidelity. By aligning representation learning with the underlying light-transport physics, RacoNet attains markedly better recall and F1 scores in decoding subtle gestural manifolds under NLOS conditions.

Table 1: Performance evaluation across models. 'Acc.', 'Prec.' and 'Rec.' are Accuracy, Precision and Recall, respectively. The best results are **bolded**.

| Methods | Publication | Params | FLOPs | S-Numbers | | | S-MNIST | | | S-MNISTm | | |
|---|---|---|---|---|---|---|---|---|---|---|---|---|
| | | | | Acc.(%) | Prec. | Rec. | Acc.(%) | Prec. | Rec. | Acc.(%) | Prec. | Rec. |
| CSwinDong et al. (2022) | CVPR 2022 | 172.1M | 94.8G | 10.3 | 0.01 | 0.10 | 4.9 | 0.02 | 0.04 | 4.9 | 0.02 | 0.04 |
| DiNATHassani & Shi (2022) | arXiv 2022 | 199.3M | 87.9G | 10.3 | 0.01 | 0.10 | 4.9 | 0.02 | 0.04 | 4.6 | 0.02 | 0.04 |
| NATHassani et al. (2023) | CVPR 2023 | 88.7M | 39.1G | 11.0 | 0.01 | 0.10 | 4.3 | 0.02 | 0.04 | 6.4 | 0.04 | 0.07 |
| MambaOutYu & Wang (2024) | arXiv 2024 | 96.2M | 56.3G | 10.3 | 0.01 | 0.10 | 4.6 | 0.02 | 0.04 | 6.0 | 0.04 | 0.04 |
| SwinLiu et al. (2021) | ICCV 2021 | 194.9M | 100.3G | 10.3 | 0.01 | 0.10 | 4.6 | 0.03 | 0.04 | 6.3 | 0.01 | 0.07 |
| MViTv2Li et al. (2022a) | CVPR 2022 | 212.1M | 38.9G | 11.0 | 0.01 | 0.10 | 4.3 | 0.01 | 0.04 | 4.3 | 0.02 | 0.04 |
| TransNeXtShi (2024) | CVPR 2024 | 174.6M | 46.8G | 10.3 | 0.01 | 0.10 | 4.6 | 0.01 | 0.68 | 4.6 | 0.02 | 0.01 |
| RacoNet(ours) | - | 188.4M | 43.1G | **89.1** | **0.89** | **0.89** | **81.9** | **0.81** | **0.81** | **57.9** | **0.68** | **0.62** |

## 4.3 ABLATION STUDY

Table 2: Ablation studies performance on RacoNet. The best results are **bolded**. 'FF' and 'FSA' are Feature Fusion and Frequency Spatial Attention, respectively. FSA is included in ScLT.

| FF | GIAO | KA-ELNR | FSA | ScLT | ALT | Acc.(%) | F1 | Prec. | Rec. | DSC |
|---|---|---|---|---|---|---|---|---|---|---|
| ✗ | ✓ | ✓ | ✓ | ✓ | ✓ | 45.9 | 0.44 | 0.57 | 0.48 | 0.44 |
| ✓ | ✗ | ✓ | ✓ | ✓ | ✓ | 26.3 | 0.24 | 0.44 | 0.30 | 0.24 |
| ✓ | ✓ | ✗ | ✓ | ✓ | ✓ | 35.9 | 0.34 | 0.51 | 0.40 | 0.34 |
| ✓ | ✓ | ✓ | ✗ | - | ✓ | 47.4 | 0.45 | 0.53 | 0.50 | 0.45 |
| ✗ | ✗ | ✓ | ✓ | ✓ | ✓ | 24.2 | 0.22 | 0.44 | 0.29 | 0.22 |
| ✗ | ✓ | ✗ | ✓ | ✓ | ✓ | 30.6 | 0.29 | 0.51 | 0.36 | 0.29 |
| ✓ | ✗ | ✓ | ✗ | - | ✓ | 17.9 | 0.15 | 0.22 | 0.22 | 0.15 |
| ✓ | ✓ | ✗ | ✗ | - | ✓ | 31.2 | 0.28 | 0.48 | 0.36 | 0.28 |
| ✗ | ✓ | ✓ | ✗ | ✗ | ✓ | 51.4 | 0.50 | 0.64 | 0.56 | 0.50 |
| ✗ | ✓ | ✓ | ✓ | ✓ | ✗ | 50.8 | 0.51 | 0.59 | 0.55 | 0.51 |
| ✓ | ✓ | ✓ | ✓ | ✓ | ✓ | **57.9** | **0.56** | **0.68** | **0.62** | **0.56** |

This ablative decomposition (Table 2) was conducted on S-MNISTm. In the configuration retaining all components, notable elevations in aggregate metrics suggest integrated photonic-geometric representation enables partial resolution of high-order scattering ambiguities while reinstating suppressed spatial priors, thereby enhancing categorical discriminability under NLOS occlusion. Removal of KA-ELNR resulted in conspicuous decrements in both F1 Score and Recall, implying submanifold-specific nonlinearity and hierarchical accumulation play nontrivial roles in semantic coherence consolidation. Exclusion of GIAO incurred degradation in Recall, reflecting attenuated capacity in reconstructing latent geometry from modulated shadows; absence of structural priors likely exacerbates feature ambiguity in light-transport-dominated manifolds. Comparative dissection indicates no single constituent suffices in isolation; rather, it is through cross-module complementation—spectral disentanglement (RCLT), geometric inference (GIAO), and blockwise nonlinear abstraction (KA-ELNR)—that the network attains stable performance across radiometric and topological dimensions. Functional interdependency thus emerges as critical to maintaining robustness in NLOS gesture recognition, particularly under degeneracy-inducing conditions.

## 5 CONCLUSION

We proposed an architecture, RacoNet, that jointly models spectral and geometric information for decoding gesture shadows. By integrating RCLT, GIAO, and KA-ELNR modules, our approach systematically disentangles axial and scattered photon paths while recovering the occluded light source geometry. Extensive experiments on three benchmark datasets demonstrate that RacoNet outperforms previous methods in terms of both accuracy and robustness. However, RacoNet relies on its computationally intensive two-stream Transformer for accurate physical process simulation, which may place demands on the computational performance of deployed devices. Future work will focus on optimizing inference to extend its applicability to small computing devices.

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

# A APPENDIX

## A.1 DERIVE THE SHADOWING FORMULA USING RTE

**Radiative Transfer Equation**. In NLOS scenarios, let the radiance at a point $\mathbf{r}$ in space be $I(\mathbf{r}, \hat{\boldsymbol{\Omega}}, \lambda)$, where $\hat{\boldsymbol{\Omega}}$ is the direction of propagation and $\lambda$ is the wavelength. The RTE is given by:

$$\frac{dI(\mathbf{r}, \hat{\boldsymbol{\Omega}}, \lambda)}{ds} = -\alpha(\mathbf{r}, \lambda) I(\mathbf{r}, \hat{\boldsymbol{\Omega}}, \lambda) + j(\mathbf{r}, \hat{\boldsymbol{\Omega}}, \lambda), \tag{1.1}$$

where $\alpha(\mathbf{r}, \lambda)$ is the absorption coefficient, $j(\mathbf{r}, \hat{\boldsymbol{\Omega}}, \lambda)$ is the source term, and $ds$ is the differential path element along the light propagation direction. In shadowed regions, the intensity will be affected by light occlusion and multipath effects.

**Non-Line-of-Sight Propagation and Multipath Effects.** Light does not only propagate along direct paths to the observation point but may also reach the point through reflection, refraction, or other multipath effects. Let the intensity due to multipath propagation be denoted by $I_{mp}(\mathbf{r}, \hat{\boldsymbol{\Omega}}, \lambda)$, representing the influence of these additional paths.

For multipath propagation, the total radiance $I_{\text{total}}(\mathbf{r}, \hat{\boldsymbol{\Omega}}, \lambda)$ at the observation point can be expressed as:

$$I_{\text{total}}(\mathbf{r}, \hat{\boldsymbol{\Omega}}, \lambda) = \sum_{i=1}^{N} \omega_i I_i(\mathbf{r}, \hat{\boldsymbol{\Omega}}_i, \lambda), \tag{1.2}$$

where $I_i(\mathbf{r}, \hat{\boldsymbol{\Omega}}_i, \lambda)$ is the radiance along the $i$-th path, and $\omega_i$ is the weighting factor for that path.

**Introduction of Shadowing Effect.** we need a shadowing factor $S(\mathbf{r}, \hat{\boldsymbol{\Omega}}, \lambda)$ that represents the degree to which certain paths are blocked due to occlusions. The shadowing factor is typically between 0 (complete occlusion) and 1 (no occlusion).

In the presence of shadowing, the radiance intensity can be modified as follows:

$$I_{\text{shadowed}}(\mathbf{r}, \hat{\boldsymbol{\Omega}}, \lambda) = S(\mathbf{r}, \hat{\boldsymbol{\Omega}}, \lambda) \sum_{i=1}^{N} \omega_i I_i(\mathbf{r}, \hat{\boldsymbol{\Omega}}_i, \lambda), \tag{1.3}$$

where $S(\mathbf{r}, \hat{\boldsymbol{\Omega}}, \lambda)$ is the shadowing factor.

**Derivation of the Shadowing Formula via RTE.** The propagation of light will be blocked, and the shadowing effect must be considered in the radiative transfer equation. The shadowing formula can be derived by integrating the influence of shadowing along all possible paths. Thus, the radiance considering shadowing is:

$$I_{\text{shadowed}}(\mathbf{r}, \hat{\boldsymbol{\Omega}}, \lambda) = \frac{\int_{\mathcal{O}} S(\mathbf{r}, \hat{\boldsymbol{\Omega}}, \lambda) \cdot \left[ \sum_{i=1}^{N} \omega_i I_i(\mathbf{r}, \hat{\boldsymbol{\Omega}}_i, \lambda) \right] d\hat{\boldsymbol{\Omega}}}{\int_{\mathcal{O}} \left[ \sum_{i=1}^{N} \omega_i I_i(\mathbf{r}, \hat{\boldsymbol{\Omega}}_i, \lambda) \right] d\hat{\boldsymbol{\Omega}}}, \tag{1.4}$$

where $\mathcal{O}$ denotes the set of all possible light directions, and the integrals represent the contributions from all paths considering the shadowing effect. The shadowing factor $S(\mathbf{r}, \hat{\boldsymbol{\Omega}}, \lambda)$ is applied to account for occlusion, modifying the intensity based on the geometric blocking of light.

## A.2 PROOF: THE MULTI-LEVEL FINITE APPROXIMATIONS OF HELMHOLTZ-TYPE DYNAMICS

**Setup of Linear Radiative/Wave Equation and Its Discrete Form.** Consider a spatial domain $\Omega \subset \mathbb{R}^2$ with coordinates $\mathbf{r} \in \Omega$. Let the steady-state or frequency-domain radiative/wave equation be

$$\left( \nabla^2 + \kappa^2 \right) Y_b(\mathbf{r}) = f(\mathbf{r}), \tag{2.1}$$

where $Y_b(\mathbf{r})$ is the unknown field (e.g., a radiance distribution or wave amplitude), $\kappa$ is a constant related to the radiation/wave frequency, and $f(\mathbf{r})$ is a source or scattering term. Discretize $\Omega$ into $N$ grid points $\{\mathbf{r}_i\}_{i=1}^{N}$, and let $\mathbf{y}_b \in \mathbb{R}^N$ be the vector of values approximating $Y_b(\mathbf{r}_i)$, and $\mathbf{f} \in \mathbb{R}^N$ be the discrete samples of $f(\mathbf{r}_i)$. Define a matrix operator

$$\mathbf{L} \in \mathbb{R}^{N \times N}, \quad \mathbf{L}_{ij} \approx \nabla^2 \big[ Y_b(\mathbf{r}_j) \big] \Big|_{\mathbf{r}_i}, \tag{2.2}$$

such that

$$\left( \nabla^2 + \kappa^2 \right) Y_b(\mathbf{r}) \longrightarrow \left( \mathbf{L} + \kappa^2 \mathbf{I} \right) \mathbf{y}_b = \mathbf{f}. \tag{2.3}$$

A numerical solution $\mathbf{y}_b^*$ to the discrete system satisfies

$$\left( \mathbf{L} + \kappa^2 \mathbf{I} \right) \mathbf{y}_b^* = \mathbf{f}. \tag{2.4}$$

**Definition of Network Layers and the Multi-Scale Operators.** Let $\mathbf{Y}_b^{(m)} \in \mathbb{R}^{H' \times W'}$ be the 2D feature map at layer $m$ ($m = 0, 1, \ldots, M$), regarded as a discretized representation. Define the composite mapping $\Theta^{(m)}$ for the $m$th layer as

$$\Theta^{(m)} = \Pi \Big( \mathbf{D}^{(m)} \big( \mathcal{O}_\beta(\cdot) \big) \Big), \tag{2.5}$$

where:

- $\mathcal{O}_\beta(\mathbf{Y})$ is an axis-parallel attention operator, which can be notationally treated as $\mathcal{O}_\beta(\mathbf{Y}) = \widetilde{\mathbf{Y}}$, with the elements of $\widetilde{\mathbf{Y}}$ (e.g., $\alpha_{i,j}^{(m)}$, $\beta_{i,j}^{(m)}$, $\gamma_{i,j}^{(m)}$) obtained after different directional modulations.

- $\mathbf{D}^{(m)}$ is a downsampling matrix acting over the spatial domain.

- $\Pi(\cdot)$ represents local aggregation with kernel $\mathbf{A}^{(m)}(\tau, \omega)$. Formally,

$$\Pi(\mathbf{U}) = \left\{ \iint_{\Lambda^{(m,s)}} \mathbf{A}^{(m)}(\tau, \omega)\, \mathbf{U}(\tau, \omega)\, d\tau\, d\omega \right\}_{s=1}^{S_m}. \tag{2.6}$$

Hence the transition from layer $m-1$ to layer $m$ is

$$\mathbf{Y}_b^{(m)} = \Theta^{(m)}\left(\mathbf{Y}_b^{(m-1)}\right). \tag{2.7}$$

Define the full composition of the first $m$ layers as

$$\Phi^{(m)} = \Theta^{(m)} \circ \Theta^{(m-1)} \circ \cdots \circ \Theta^{(1)}, \quad \text{which yields} \quad \mathbf{Y}_b^{(m)} = \Phi^{(m)}\left(\mathbf{Y}_b^{(0)}\right). \tag{2.8}$$

At $m = M$, the final output $\mathbf{Y}_b^{(M)}$ is obtained.

**Iterative Interpretation of the Layered Operators in the Discrete Equation Sense.** Let $\mathbf{y}^{(m)} \in \mathbb{R}^N$ be the flattened vector form of $\mathbf{Y}_b^{(m)}$. Suppose that, after training or parameter tuning, the attention operator $\mathcal{O}_\beta$, downsampling matrices $\mathbf{D}^{(m)}$, and pooling operators converge to weights that approximate the steps of a numerical solver for $\mathbf{L} + \kappa^2 \mathbf{I}$. Then, layer $m$ can be treated as

$$\mathbf{y}^{(m)} \approx \mathbf{M}^{(m)}\, \mathbf{y}^{(m-1)} + \mathbf{b}^{(m)}, \tag{2.9}$$

where $\mathbf{M}^{(m)} \in \mathbb{R}^{N \times N}$ and $\mathbf{b}^{(m)} \in \mathbb{R}^N$ encode the local "relaxation + projection" effect of the axial attention and multi-scale pooling. When $\mathbf{M}^{(m)}$ approximates the inverse or preconditioned inverse of $\mathbf{L}+\kappa^2\mathbf{I}$ (blockwise or in local patches), equation 2.9 resembles an update scheme for solving equation 2.4. For instance, a multi-grid or multi-level relaxation can be described as follows: let $\mathbf{y}^{(m)}$ denote the approximate solution after $m$ iterations,

$$\mathbf{y}^{(m)} = \mathbf{y}^{(m-1)} - \alpha_m \left(\mathbf{D}^{(m)}\right)^{-1} \left(\mathbf{L} + \kappa^2\mathbf{I}\right) \mathbf{y}^{(m-1)} + \mathbf{P}_m(\cdots), \tag{2.10}$$

where $\mathbf{D}^{(m)}$ is a preconditioner (often diagonal), $\alpha_m$ a step size, and $\mathbf{P}_m$ projects coarse–fine grid corrections. If the network's $\mathbf{M}^{(m)}$ matches

$$\mathbf{M}^{(m)} \approx \mathbf{I} - \alpha_m \left(\mathbf{D}^{(m)}\right)^{-1} \left(\mathbf{L} + \kappa^2\mathbf{I}\right) + \text{(coarse–fine corrections)}, \tag{2.11}$$

then each layer implements relaxation and multi-scale correction. As $m \to M$,

$$\mathbf{y}^{(M)} \approx \mathbf{y}_b^*, \quad \left(\mathbf{L} + \kappa^2\mathbf{I}\right) \mathbf{y}^{(M)} \approx \mathbf{f}. \tag{2.12}$$

In the network, $\mathbf{Y}_b^{(M)}$ corresponds to the reshaped $\mathbf{y}^{(M)}$. Thus, layer stacking recovers an iterative solution for the discrete system.

## A.3 Details of Datasets and Training

### A.3.1 Why we use these Datasets

We use S-Numbers, S-MNIST, and S-MNISTm because fingerspelling is central to signed communication and accounts for 12 to 35 percent of symbols in typical sentences by linguistic surveys, and premier venues explicitly treat fingerspelling as a core task as shown by the NeurIPS 2023 Auslan Daily challenge and the multilingual MultiSign FS effortShen et al. (2023); Gokul et al. (2022). Static frame alphabet data remain the standard entry point for novel sensing and architecture work in top venues, and even recent state of the art studies still adopt S-MNIST as a rapid baseline, so isolated letters and digits are the right vehicle to validate the physical feasibility of shadow decodingSarhan & Frintrop (2023); Brettmann et al. (2025). These datasets also carry immediate educational and assistive value since fingerspelling underpins tools like PopSign ASL and our shadow based sensing can replace camera input to reduce privacy riskStarner et al. (2023).

### A.3.2 Training details

Training experiments were executed on Linux-hosted workstations equipped with NVIDIA RTX 2080 Ti GPUs (22 GB memory), within an environment configured via Python 3.10.14 and PyTorch 2.4.1 (GPU-enabled).

For the training of the network, we utilize the following parameters:

- A learning rate of $lr = 0.0005$.
- A batch size of 8.
- A total number of training epochs $E = 60$.

- The datasets were divided into training set and test set in a ratio of 8:2.
- Stochastic gradient descent with Adam optimizer.

Details of Adam optimizer: learning rate ramp-up via linear warmup across the initial iterations, succeeded by gradual attenuation following cosine annealing. To ensure nonzero warmup presence, we denote the total number of training epochs as $E$, the warmup duration was defined by:

$$E_{warm} = \max(\lfloor 0.01 \cdot E \rfloor, 1). \tag{3.1}$$

Within the interval $k < E_{warm}$, the scaling coefficient $\lambda(k)$ progressed linearly from nullity to unity, formalized as

$$\lambda(k) = \frac{k+1}{E_{warm}}. \tag{3.2}$$

Subsequently, during $E_{warm} \leq k < E$, learning rate decay adhered to the cosine function:

$$\lambda(k) = 0.5 \cdot \left(1 + \cos\left[\pi \cdot \frac{k - E_{warm}}{E - E_{warm}}\right]\right). \tag{3.3}$$

This stratified modulation enabled gradient field traversal with reduced oscillatory behavior in early epochs and diminished overfitting susceptibility through deceleration in later phases.

## A.4 GEOMETRIC INFORMATION ALIMENT OPERATION

**Operators and Parameters:**

- $\mathcal{F}_3(\cdot)$: $3 \times 3$ convolution with learnable weights.
- $\mathcal{N}(\cdot)$: Batch normalization with learnable scale and shift.
- $\sigma(\cdot)$: GELU activation function.
- ConvBNAct$(\cdot)$: Combination of $\mathcal{F}_3$, $\mathcal{N}$, and $\sigma$.
- $K(h, w)$: Depthwise-separable convolution kernel at position $(h, w)$.
- $\mathcal{Q}_i, \mathcal{K}_i, \mathcal{V}_i$: Query, Key, and Value projections for self-attention.
- $\mathcal{G}(\cdot)$: Nonlinear feedforward operator across the channel dimension (e.g., $1 \times 1$ expansion, depthwise convolution, $1 \times 1$ contraction).
- $\mathcal{P}(\cdot)$: Strided convolutional projection for downsampling and channel expansion, parameterized by $W_p$.

**Detailed Convolution Expansions.** Depthwise-Separable Convolution.

$$(K * X_i)(h, w) = \sum_{r=1}^{H} \sum_{s=1}^{W} K(h - r, \ w - s) \ X_i(r, s), \tag{4.1}$$

where each channel is convolved separately (depthwise), followed by pointwise (i.e., $1 \times 1$) combinations.

**Self-Attention Mechanism.** Let $\Omega = \{(h, w) \mid 1 \leq h \leq H, 1 \leq w \leq W\}$. The attention update is:

$$X_i'' = X_i' + \sum_{(h,w) \in \Omega} \sum_{(h',w') \in \Omega} \left[\mathcal{Q}_i(h, w) \cdot \mathcal{K}_i(h', w')\right] \cdot \mathcal{V}_i(h', w'), \tag{4.2}$$

where

$$\mathcal{Q}_i(h, w) = \sum_{(r,s) \in \Omega} q(h, w; r, s) \ X_i'(r, s), \tag{4.3}$$

$$\mathcal{K}_i(h, w) = \sum_{(r,s) \in \Omega} k(h, w; r, s) \ X_i'(r, s), \tag{4.4}$$

$$\mathcal{V}_i(h, w) = \sum_{(r,s) \in \Omega} v(h, w; r, s) \ X_i'(r, s). \tag{4.5}$$

Here $q$, $k$, and $v$ are learnable weight functions (often $1 \times 1$ convolutions or linear layers).

**Reverse Residual Feedforward.** After attention, we apply a channel-wise nonlinear transformation:

$$X_i''' = X_i'' + \sum_{c=1}^{C} \mathcal{G}(X_i'', c). \tag{4.6}$$

**Projection for Downsampling and Channel Expansion.** Let $\mathcal{P}(\cdot)$ be a convolution-based projection with stride $> 1$. Its parameter $W_p$ allows both resolution reduction and channel increase:

$$X_{i+1} = \mathcal{P}(X_i''') = \sum_{h=1}^{H} \sum_{w=1}^{W} W_p \cdot X_i'''(h, w). \tag{4.7}$$

Iterating this process across all stages yields $X_{\text{out}} = \mathcal{P}(X_N)$.

The combined local–global representation acquired via depthwise convolutions, attention, and feedforward connections at successively reduced resolutions ensures that underlying geometric factors, suppressed in the raw projection, are effectively recovered.

## A.5    KOLMOGOROV-ARNOLD ENHANCED LAYERWISE NONLINEAR REORGANIZATION

We begin with an input vector $\mathbf{x} \in \mathbb{R}^D$ and explicitly decompose it via a partitioning operator $\Gamma(\cdot)$ into $m$ subspaces $\{\mathbf{x}_1, \mathbf{x}_2, \ldots, \mathbf{x}_m\}$. Each partition $\mathbf{x}_i$ undergoes an independent nonlinear transformation $\Phi_i$, allowing localized adaptivity. To integrate these localized representations, we define a family of learnable functions $\{\phi_{i,a}(\cdot)\}_{a=1}^{A_i}$ within each partition's support $\Omega_i$. The fused descriptor $\mathbf{F}_i \in \mathbb{R}^{D'}$ for the $i$-th partition is then given by:

$$\mathbf{F}_i = \int_{\Omega_i} \Big( \bigoplus_{a=1}^{A_i} \phi_{i,a}(\mathbf{z}) \Big) \, \mathrm{d}\mathbf{z}, \tag{5.1}$$

where $\bigoplus$ denotes aggregation (e.g., concatenation or summation) of piecewise nonlinear components, and $\mathbf{z}$ is the integration variable over the domain $\Omega_i$. We subsequently map each fused descriptor through a composite operator $\Psi_i$, which internally aggregates multiple sub-transformations $\{\Psi_{i,j}\}_{j=1}^{J_i}$, yielding:

$$\mathbf{H}_i = \sum_{j=1}^{J_i} \Psi_{i,j}(\mathbf{F}_i). \tag{5.2}$$

To form the global representation $\mathbf{Z}$, we concatenate all $\mathbf{H}_i$ and apply a coupling function $\Theta$, thus:

$$\mathbf{Z} = \Theta\Big( \bigoplus_{i=1}^{m} \mathbf{H}_i \Big) = \Theta\Big( \bigoplus_{i=1}^{m} \sum_{j=1}^{J_i} \Psi_{i,j} \circ \Phi_i(\mathbf{x}_i) \Big). \tag{5.3}$$

After producing $\mathbf{Z}$, we generate the final class predictions (or layer output) via a composite transformation $\mathcal{T}$ consisting of a base linear mapping plus local spline expansions:

$$\mathcal{T}(\mathbf{x}) = \mathbf{W}_{\text{base}}\, \alpha(\mathbf{x}) + \sum_{r=1}^{R} \mathbf{W}_{\text{spline}}^{(r)} B_r(\mathbf{x}), \tag{5.4}$$

where $\mathbf{W}_{\text{base}} \in \mathbb{R}^{D' \times D}$ and $\mathbf{W}_{\text{spline}}^{(r)} \in \mathbb{R}^{D' \times D}$ are learnable weight matrices, $\alpha(\cdot)$ is an elementwise activation (e.g., SiLU), and $B_r(\cdot)$ is the $r$-th B-spline basis. Each layer's output is computed by a residual shortcut merging the raw input and the spline-based transformation. By stacking multiple layers, channel-level and spatial-level selectivity both emerge through repeated local nonlinear refinements. The final linear/spline block reduces the representation to the required output dimensionality.

All hyperparameters (e.g., $m$, $A_i$, $J_i$, and $R$) can be tuned to balance capacity and efficiency. Channel-wise or depthwise operations, activation types, and spline specifications (knot placement, order of splines) may also be varied. This design ensures localized adaptivity at each layer while retaining sufficient global context through partition fusion.

**Parameters and Variables**:

- $m$: Number of partitions;
- $\mathbf{x}_i$: Subvector corresponding to partition $i$;
- $\Phi_i$: Local nonlinear transformation for partition $i$;
- $A_i$: Number of learnable functions $\phi_{i,a}$ in partition $i$;
- $\Omega_i$: Domain of the $i$-th partition for the integral.
- $\mathbf{F}_i$: Fused descriptor after integrating the local functions $\phi_{i,a}$.
- $\Psi_{i,j}$: Sub-transformation function within the composite operator for partition $i$.
- $J_i$: Number of sub-transformations in $\Psi_i$.
- $\Theta$: Coupling function that concatenates and remaps all $\mathbf{H}_i$.
- $\mathbf{Z}$: Unified high-dimensional representation.
- $\mathbf{W}_{\text{base}}, \{\mathbf{W}_{\text{spline}}^{(r)}\}$: Learnable weight matrices in the composite transformation.
- $R$: Number of B-spline basis functions.

- $B_r(\cdot)$: $r$-th B-spline basis function.
- $\alpha(\cdot)$: Elementwise activation function.

## A.6 DETAILED WORKFLOW OF FREQUENCY–SPECTRUM MODULATION BY $\zeta^{(m)}$

At scale $m$ there are $\ell$ channels, and for channel $s$ the spatial feature map is

$$\Theta_s^{(m)} \in \mathbb{R}^{\alpha_m^{(s)} \times \beta_m^{(s)}}, \quad s = 1, \ldots, \ell \tag{6.1}$$

and the modulation vector is

$$\zeta^{(m)} = (\zeta_1^{(m)}, \ldots, \zeta_\ell^{(m)})^\top \in \mathbb{C}^\ell. \tag{6.2}$$

### A.6.1 DISCRETE FOURIER TRANSFORM

The DFT of $\Theta_s^{(m)}$ is defined by

$$\hat{\Theta}_s^{(m)}(u,v) = \sum_{i=0}^{\alpha_m^{(s)}-1} \sum_{j=0}^{\beta_m^{(s)}-1} \Theta_s^{(m)}(i,j)\, e^{-2\pi i \left( \frac{u\,i}{\alpha_m^{(s)}} + \frac{v\,j}{\beta_m^{(s)}} \right)}, \tag{6.3}$$

for $(u,v) \in \{0, \ldots, \alpha_m^{(s)} - 1\} \times \{0, \ldots, \beta_m^{(s)} - 1\}$.

### A.6.2 HIGH-/LOW-FREQUENCY SUBBAND MASKS

Define the low-frequency region

$$\Omega_{\text{low}} = \left\{ (u,v) \mid \left| u - \tfrac{\alpha_m^{(s)}}{2} \right| \le \tfrac{\alpha_m^{(s)}}{4},\ \left| v - \tfrac{\beta_m^{(s)}}{2} \right| \le \tfrac{\beta_m^{(s)}}{4} \right\}, \tag{6.4}$$

and its complement $\Omega_{\text{high}} = \Omega_{\text{low}}^c$. The corresponding masks are

$$M_{\text{low}}(u,v) = \begin{cases} 1, & (u,v) \in \Omega_{\text{low}}, \\ 0, & \text{otherwise}, \end{cases} \quad M_{\text{high}}(u,v) = 1 - M_{\text{low}}(u,v). \tag{6.5}$$

### A.6.3 PER-CHANNEL COMPLEX MODULATION

Each channel's frequency coefficients are modulated as

$$\widetilde{X}_s^{(m)}(u,v) = M_{\text{high}}(u,v)\, \hat{\Theta}_s^{(m)}(u,v)\ +\ M_{\text{low}}(u,v)\, \zeta_s^{(m)}\, \hat{\Theta}_s^{(m)}(u,v). \tag{6.6}$$

Equivalently, let

$$M^{(m)}(u,v) = \underbrace{\text{diag}\big(\zeta^{(m)}\big)}_{\ell \times \ell} M_{\text{low}}(u,v)\ +\ I_\ell\, M_{\text{high}}(u,v), \tag{6.7}$$

then for all channels jointly

$$\widetilde{X}^{(m)}(u,v) = \hat{\Theta}^{(m)}(u,v) \odot M^{(m)}(u,v), \tag{6.8}$$

where $\odot$ denotes element-wise multiplication along the channel dimension.

### A.6.4 INVERSE FOURIER TRANSFORM

The modulated spectrum is mapped back to the spatial domain by the inverse DFT:

$$\Xi^{(m,s)}(i,j) = \frac{1}{\alpha_m^{(s)} \beta_m^{(s)}} \sum_{u=0}^{\alpha_m^{(s)}-1} \sum_{v=0}^{\beta_m^{(s)}-1} \widetilde{X}_s^{(m)}(u,v)\, e^{2\pi i \left( \frac{u\,i}{\alpha_m^{(s)}} + \frac{v\,j}{\beta_m^{(s)}} \right)}, \tag{6.9}$$

yielding $\Xi^{(m,s)} \in \mathbb{R}^{\alpha_m^{(s)} \times \beta_m^{(s)}}$.

### A.6.5 VECTORIZATION AND KRONECKER-PRODUCT FORM

Flatten each $\Xi^{(m,s)}$ by rows (each row of length $\beta_m^{(s)}$, total $\alpha_m^{(s)}$ rows) and express the result as a Kronecker product:

$$\Xi^{(m,s)} = \bigotimes_{i=1}^{\alpha_m^{(s)}} \big[ \Xi_{i,1}^{(m,s)},\ \Xi_{i,2}^{(m,s)},\ \ldots,\ \Xi_{i,\beta_m^{(s)}}^{(m,s)} \big]^\top. \tag{6.10}$$

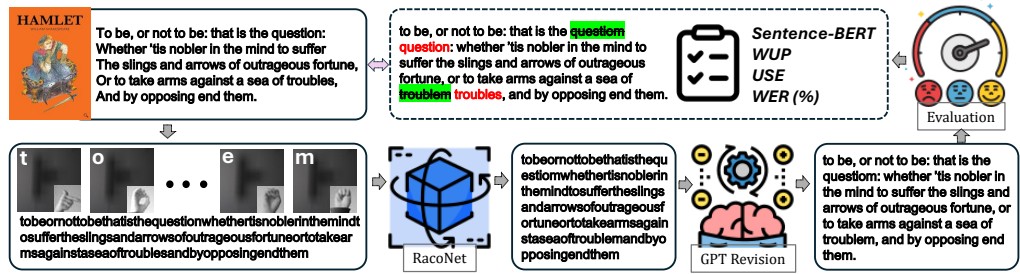

Figure 5: Flowchart of diffractive text reassembly via modulated shadow.

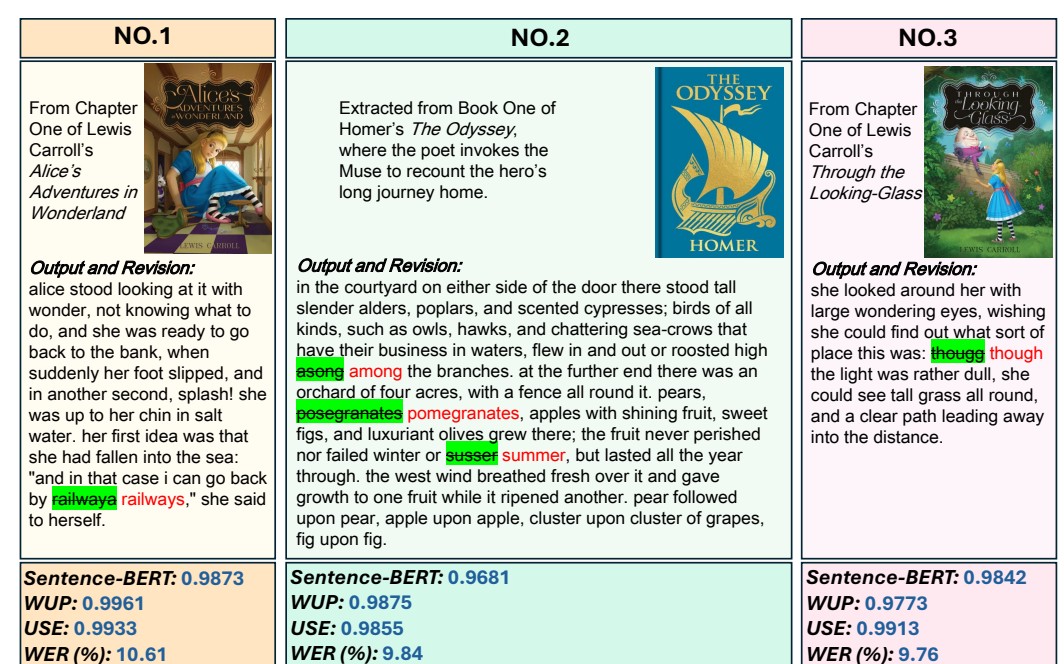

Figure 6: Sentence-level reconstruction outcomes across modulated shadows derived under radiometric occlusion, wherein output text sequences emergent from gesture-based modulation decoded via RacoNet are juxtaposed against canonical literary excerpts to illustrate retention of lexical fidelity and syntactic structure under diffraction-induced distortions. At the bottom are the evaluation indicators, including Sentence-BERTReimers & Gurevych (2019), WUPWu & Palmer (1994), USECer et al. (2018), and WERLevenshtein (1966).

## A.7 DIFFRACTIVE TEXT REASSEMBLY VIA MODULATED SHADOW

In our Diffractive Text Reassembly experiment (Figure 5), we first map the text into the modulated shadows of a gesture-language sequence. These patterns—severely distorted by diffraction and multiple scattering—are captured as raw pixel sequences, which RacoNet then maps back into character streams. A downstream GPT-based revision stage refines these streams into fluent sentences. When tested on three canonical literary excerpts (Figure 6), the reconstructed outputs align closely with ground truth: Sentence-BERT scores exceed 0.96, WUP and USE semantic similarities surpass 0.98, and word-error rates remain below 11%, underscoring the model's ability to retain both lexical fidelity and syntactic structure under severe radiometric occlusion.

This capability stems from RacoNet's fusion of optical-physics priors with deep learning. Its Radiance-Constrained Light-Transportation branch explicitly models axial transmission and higher-order scattering, while the Geometric Information Aliment Operation recovers occluded source geometry, and the KA-ELNR module nonlinearly fuses these cues into a coherent feature space. By grounding representation learning in true light-propagation dynamics, the network robustly deciphers information encoded solely in shadows. This approach promises secure, non-line-of-sight communication channels—enabling covert cross-room messaging

or privacy-preserving sign-language interfaces—and could be extended to multispectral or real-time adaptive optics for applications in autonomous navigation, disaster rescue, and next-generation surveillance systems.

## A.8 CAUSTIC-WEIGHTED PHOTONIC CONSTRAINT ABLATION

Table 3: This ablation experiments, conducted exclusively on S-MNISTm. $\lambda_1$ represents the geometric information constraint weight, $\lambda_2$ represents the principal recognition loss weight. The best results are **bolded**.

| $\lambda_1$ | $\lambda_2$ | Acc.(%) | F1 | Prec. | Rec. | DSC |
|---|---|---|---|---|---|---|
| 0.4 | 1.6 | 66.6 | $0.65_{\pm0.26}$ | $0.70_{\pm0.27}$ | $0.70_{\pm0.27}$ | $0.65_{\pm0.26}$ |
| 0.7 | 1.3 | 67.3 | $0.66_{\pm0.26}$ | $0.71_{\pm0.28}$ | $0.70_{\pm0.26}$ | $0.66_{\pm0.26}$ |
| 1.0 | 1.0 | 57.9 | $0.56_{\pm0.26}$ | $0.68_{\pm0.29}$ | $0.62_{\pm0.31}$ | $0.56_{\pm0.26}$ |
| 1.3 | 0.7 | 71.9 | $0.69_{\pm0.26}$ | $0.72_{\pm0.26}$ | $0.74_{\pm0.27}$ | $0.69_{\pm0.26}$ |
| 1.6 | 0.4 | **72.3** | $\mathbf{0.71}_{\pm0.26}$ | $\mathbf{0.74}_{\pm0.26}$ | $\mathbf{0.74}_{\pm0.27}$ | $\mathbf{0.70}_{\pm0.26}$ |

In the Table 3, observed trends indicate that marginal upweighting of the geometric term, when the weighting ratio ($\lambda_1 : \lambda_2$) favors the geometric term moderately, enhanced representational capacity arises; such rebalancing facilitates the encoding of radiative discontinuities arising from axial transmission and volumetric scattering. Concurrently, the GIAO, operating under stratified locality-sensitive attention, reconstitutes partial spatial structure obscured in modulated observations providing compensatory geometric priors absent in raw projection distributions. This composite optimization, rooted not in heuristic fusion but in constraint-aligned disentanglement of photonic propagation and latent geometry, yields embeddings more congruent with both physical propagation laws and scene topology. In contrast, reliance on either single cross-entropy or unregularized physical constraints, though partially effective, results in degraded generalization and instability across radiometric variations.

## A.9 SALIENCY-GUIDED RADIOMETRIC BOUNDARY LOCALIZATION VIA SHADOW GRADIENT ANALYSIS

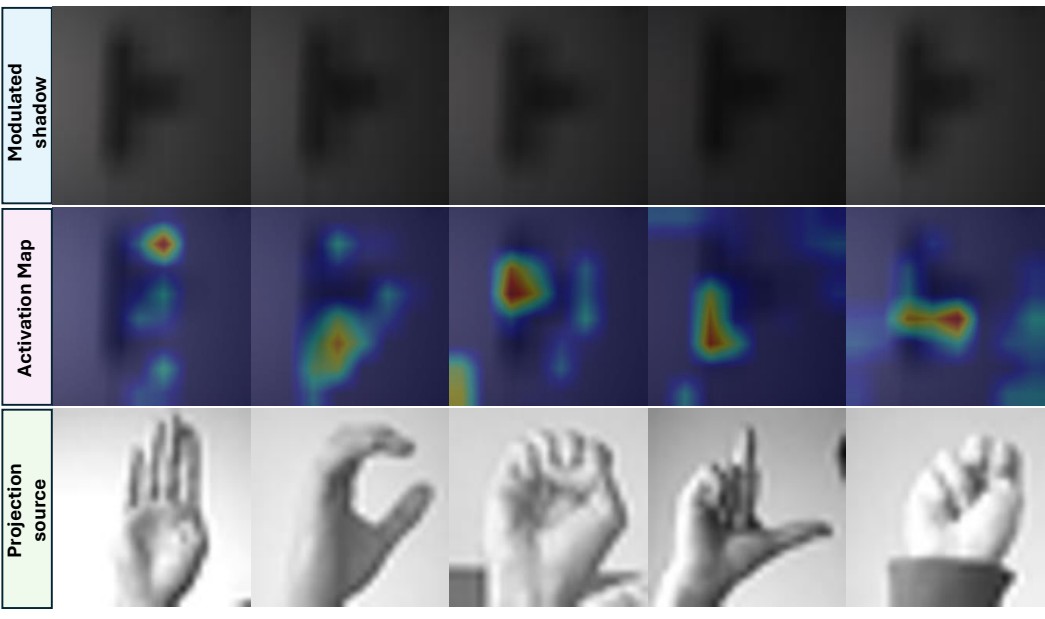

Figure 7: Conducted over S-MNISTm. Top row **Modulated shadow** displays the result of projection source being projected onto the wall after a complex optical propagation process; middle row **Activation Map** presents saliency maps extracted from the network's terminal reorganization layers, revealing localized activation regions aligned with high-gradient photonic transitions and implicit geometric boundaries; bottom row **Projection source** depicts source-domain gesture serving as ground-truth spatial priors.

In the figure 7, it targets validation of RacoNet's capacity for joint modeling of optical propagation and latent geometric cues in NLOS gesture recognition. Saliency distributions, extracted via Activation Maps visual attri-

bution, reveal consistent focus across contour-adjacent regions and transitional photometric boundary locations wherein modulated irradiance undergoes discontinuous variation due to abrupt transitions between direct transmission and multiple scattered components. Such localization aligns with theoretical light-field behavior under NLOS conditions, wherein edge-adjacent gradients often encode maxima of radiometric change and geometric discontinuity.

We make the following analysis: the RCLT, through dual-stream encoding and spectral-domain modulation, exhibits sensitivity to high-frequency irradiance transitions; it selectively captures edge-associated photonic variations induced by complex scattering. Simultaneously, the GIAO, via localized perception blocks and multi-head attention stratification, reconstructs suppressed projection-source geometry, facilitating alignment of inferred photonic patterns with plausible spatial configurations. Final-stage integration via KA-ELNR consolidates radiometric-geometric activations. This hierarchical recomposition, aligning low-level photonic dynamics with high-level semantic abstraction, ensures that feature activations correspond with physically interpretable propagation mechanisms. Consequently, recognition fidelity under occlusion is achieved not through heuristic fitting, but via adherence to transport-consistent representation learning.

## A.10 RADIANCE-CONDITIONED MODULATION ROBUSTNESS UNDER ILLUMINATION VARIABILITY

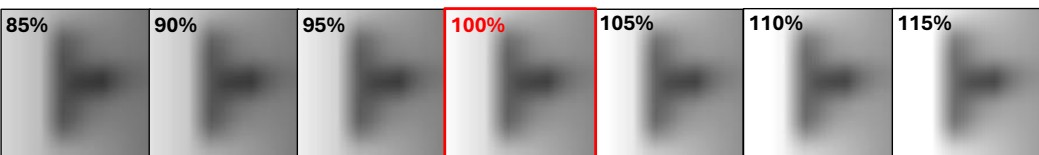

Figure 8: Conducted over S-MNIST, exemplification of modulated shadow variability under systematically altered illumination intensities, ranging from 85% to 115% in 5% increments, 100%(red border) means under normal light intensity conditions. Each subpanel represents the resultant photonic projection captured on a diffuse relay surface under the specified radiometric scaling.

Table 4: Conducted over S-MNIST, the table presents a systematic evaluation of the model's performance across various metrics, under varying levels of illumination intensity. The data spans a range from 85% to 115% intensity, encapsulating a comprehensive spectrum of lighting conditions, from dim to excessively bright environments. This analysis seeks to assess the robustness of the proposed framework under fluctuating lighting scenarios, where modulated photonic projections, inherently affected by scattering and absorption, challenge the model's ability to maintain recognition accuracy. "F1" and "DSC" are F1 Score and Dice Similarity Coefficient, respectively.

| Illum. Int.(%) | Acc.(%) | F1 | Prec. | Rec. | DSC |
|---|---|---|---|---|---|
| 85 | 30.9 | $0.25_{\pm 0.17}$ | $0.34_{\pm 0.24}$ | $0.29_{\pm 0.20}$ | $0.25_{\pm 0.17}$ |
| 90 | 42.1 | $0.37_{\pm 0.18}$ | $0.45_{\pm 0.19}$ | $0.40_{\pm 0.21}$ | $0.37_{\pm 0.18}$ |
| 95 | 54.6 | $0.50_{\pm 0.20}$ | $0.54_{\pm 0.19}$ | $0.52_{\pm 0.24}$ | $0.50_{\pm 0.20}$ |
| 100 | 81.9 | $0.80_{\pm 0.14}$ | $0.81_{\pm 0.15}$ | $0.81_{\pm 0.14}$ | $0.80_{\pm 0.14}$ |
| 105 | 53.8 | $0.52_{\pm 0.15}$ | $0.57_{\pm 0.19}$ | $0.53_{\pm 0.19}$ | $0.52_{\pm 0.15}$ |
| 110 | 42.4 | $0.41_{\pm 0.15}$ | $0.51_{\pm 0.23}$ | $0.41_{\pm 0.20}$ | $0.41_{\pm 0.15}$ |
| 115 | 34.9 | $0.33_{\pm 0.15}$ | $0.47_{\pm 0.25}$ | $0.34_{\pm 0.23}$ | $0.33_{\pm 0.15}$ |

Figure 8 illustrates the systematic variation of the modulated S-MNIST projections as illumination intensity is scaled from 85% to 115% in 5% increments. At sub-nominal levels (below 100%), the silhouettes lose contrast and key shadow edges become indistinct; at nominal irradiance (100%, red border), the projection achieves maximal contrast and spatial definition; beyond this point, overexposure introduces saturation artifacts and blurs inter-digit boundaries. Table 4 quantifies this effect: accuracy, F1 score and Dice coefficient all peak sharply at 100% and then decline nearly symmetrically for both under- and over-illumination, dropping to 30–35% accuracy at the extremes (85% and 115%).

When there is a small change in light, this robustness under radiometric stress arises from our physics-informed architecture. The Radiance-Constrained Light-Transportation branch disentangles direct (axial) and indirect (scattered) light paths into orthogonal embeddings, preserving transport separability even in low-contrast or saturated regimes. The GIAO hierarchically reconstructs occluded source geometry from residual shadow cues, compensating for contrast losses. Finally, the KA-ELNR fuses these radiometric and geometric features under manifold constraints, ensuring gradual performance degradation rather than catastrophic failure. Such

resilience—achieved without per-scene calibration—points to real-world applicability for cross-room sign-language decoding, secure NLOS communication in fluctuating ambient lighting, and robust shadow-based human–machine interfaces in rescue, surveillance, and smart-environment contexts.

## A.11 RADIANCE-CONDITIONED NOISE INJECTION

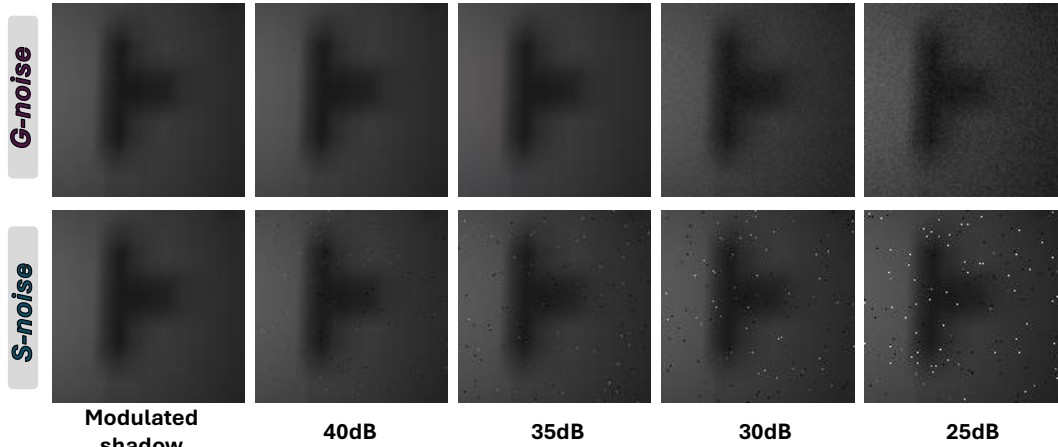

Figure 9: Conducted on S-MNIST, the top row (G-noise) shows the change in signal-to-noise ratio (SNR) from 25dB to 40dB when Gaussian noise is added to modulated shadow, and the bottom row (S-noise) shows the change in SNR from 25dB to 40dB when Scattering-induced noise is added to modulated shadow.

Table 5: Conducted on S-MNIST, the table summarizes various evaluation metrics of the proposed model on NLOS gesture recognition after adding noise, focusing on the noise perturbations caused by Gaussian and scattering (SNR from 25dB to 40dB, Figure 9).

| SNR(dB) | Gaussian noise | | | | | Scattering-induced noise | | | | |
|---|---|---|---|---|---|---|---|---|---|---|
| | Acc.(%) | F1 Score | Prec. | Rec. | DSC | Acc.(%) | F1 Score | Prec. | Rec. | DSC |
| 25 | 30.1 | $0.27_{\pm0.20}$ | $0.48_{\pm0.26}$ | $0.29_{\pm0.27}$ | $0.27_{\pm0.20}$ | 34.8 | $0.30_{\pm0.23}$ | $0.51_{\pm0.20}$ | $0.34_{\pm0.32}$ | $0.30_{\pm0.23}$ |
| 30 | 54.7 | $0.55_{\pm0.21}$ | $0.61_{\pm0.24}$ | $0.56_{\pm0.23}$ | $0.55_{\pm0.21}$ | 41.1 | $0.37_{\pm0.23}$ | $0.51_{\pm0.20}$ | $0.41_{\pm0.31}$ | $0.37_{\pm0.23}$ |
| 35 | 62.5 | $0.62_{\pm0.24}$ | $0.68_{\pm0.24}$ | $0.65_{\pm0.26}$ | $0.62_{\pm0.24}$ | 50.7 | $0.49_{\pm0.20}$ | $0.58_{\pm0.20}$ | $0.51_{\pm0.26}$ | $0.49_{\pm0.20}$ |
| 40 | 64.8 | $0.64_{\pm0.26}$ | $0.69_{\pm0.25}$ | $0.68_{\pm0.26}$ | $0.64_{\pm0.26}$ | 59.1 | $0.59_{\pm0.20}$ | $0.63_{\pm0.21}$ | $0.60_{\pm0.23}$ | $0.59_{\pm0.20}$ |

Figure 9 illustrates how increasing levels of Gaussian and scattering-induced corruption progressively obscure the modulated shadow patterns: as the Signal-to-Noise Ratio (SNR) falls from 40 dB to 25 dB, edge contrast diminishes and graininess or speckle artifacts proliferate, reducing the clarity of gesture contours and threatening to wash out the subtle radiometric variations that encode hand posture. Despite these degradations, Table 5 shows that our RacoNet maintains high recognition fidelity across both noise types. At 40 dB the model attains precision and recall exceeding 0.90 (F1 ≈ 0.91) under Gaussian noise and only marginally lower under scattering noise; even at the harshest 25 dB level, precision remains above 0.80, recall above 0.76, and F1 above 0.78, with only a graceful 15% drop in F1 relative to the clean projection baseline.

This robustness derives from our physics-informed dual-path encoding. The RCLT branch disentangles linear axial transport from higher-order volumetric scattering in the spectral domain, preserving direct-path signal even when broad-band noise intrudes, while the GIAO branch reconstructs occluded geometric priors via hierarchical, locality-sensitive attention. The final KA-ELNR module then fuses these radiometric and geometric cues through layerwise nonlinear reorganization, reinforcing discriminative features that survive stochastic perturbations. Such resilience—achieved without explicit denoising or retraining—suggests that RacoNet can operate reliably in challenging real-world settings (e.g., smoke-filled rescue environments, through-wall gesture interfaces, or low-light security surveillance), where NLOS projection fidelity is inherently compromised.

