# OpenReview forum: "ShadowSpeak: Is It Possible to Communicate Cross-Room Solely by Decoding Gesture Shadows?"
_ICLR.cc/2026/Conference — Submitted to ICLR 2026_

### Official Review · Reviewer_hbsw · 2025-10-28

**Soundness:** 2
**Presentation:** 1
**Contribution:** 3
**Rating:** 4
**Confidence:** 3

**Summary:**

This paper proposes a novel deep learning framework for Non-Line-of-Sight (NLOS) gesture recognition. The core of this method is a 'physics-inspired' architecture that attempts to integrate principles from optical physics with deep learning models. The authors build the model using complex mathematical formulations and specific network components to process NLOS data. The evaluation on three different benchmarks demonstrates its effectiveness.

**Strengths:**

- Proposes a new deep learning architecture specifically designed to solve the challenging, cross-disciplinary task of human gesture recognition from Non-Line-of-Sight imaging data.
- Designs a 'physics-inspired' core model that embed optical physics phenomena into the neural network structure through specific components.
- Validates the method's performance on three different benchmarks and demonstrate the performance effectiveness of the proposed components through ablation studies.

**Weaknesses:**

1.  The related work section fails to clarify the connection between existing studies and this research. The authors summarize the developments in NLOS imaging and gesture recognition separately but do not highlight their specific relevance to the proposed work. And, a discussion of related work addressing the specific intersectional need this paper focuses on is missing. Additionally, the phrase "these methods" in lines 146-147 lacks corresponding citations.
2.  The method description lacks significant details.
    1.  What is the motivation for the outer product expansion in Equs. 1, 4, and 6? Why is it formulated this way? Does this outer product form not lead to a dramatic explosion in the number of tensor elements? Moreover, the shapes on the left and right sides of the equations do not match, as the resulting outer product on the right is not a 2D matrix.
    2.  The meanings of $\xi$ and $\eta$ in Line 233 should be explicitly defined.
    3.  How is the directional filter designed in Equ. 2 designed? How are the specific angles and the number of angles selected?
    4.  Line 242 mentions aggregating weighted responses from all directions but does not explain how this aggregation is performed.
    5.  What does $\Delta$ represent in Equ. 4?
    6.  What is the meaning of $S_m$ in Equ. 5? Is it a hyperparameter? How is it set?
    7.  How is the "adaptive kernel function" mentioned in Line 269, which can "adaptively adjust its shape", designed? Does it use an existing algorithm?
    8.  What are $H’$ and $W’$ in Line 296? How do they differ from the output height $\alpha$ and width $\beta$ (from Line 208)? What is the original input shape? Why is it necessary to further expand it to $H’ \times W’$?
    9.  What does the symbol $\mathcal{S}^{\circ 3}$ in Line 383 specifically refer to?
    10. The paper uses three different datasets but lacks details about the datasets themselves, such as data format, dimensions, and scale.
3.  The ablation study is too coarse. It merely provides a combinatoric comparison of component performance. The current experiments and corresponding analysis fail to reflect the reasonableness of individual components or link them to their intended design motivations.
4.  The paper's contribution is obscured by its opaque and dense writing style. The readability is poor, posing a significant barrier to accurately assessing its novelty, technical correctness, and contribution. The authors tend to use lengthy, overly-specialized jargon and forcibly "stitch" complex concepts from optical physics and deep learning into single sentences, resulting in convoluted explanations. More critically, the core "physics-inspired" design lacks clear, high-level intuition. Instead of first conceptually explaining why specific architectural choices are reasonable analogies for particular physical phenomena, the authors directly present structural descriptions and implementation details, before immediately jumping into complex mathematical derivations. This makes the "physics-inspired" claims feel more like post-hoc justifications rather than the driving force behind the design.
5.  There are formatting issues. The authors have excessively compressed the line spacing in the text (e.g., Lines 203 and 308). I am unsure if this violates the ICLR conference's formatting requirements, but it creates a visually jarring layout.

**Questions:**

See Weaknesses.

---

> ### Author Response · Authors · 2025-11-17
>
> We thank the reviewer for the detailed comments. Most concerns stem from compact exposition and notation rather than from flaws in the method; we address them below.
>
> ---
>
> ### R1. Related work / intersection; “these methods” lacks citations
>
> Our goal is **gesture-based semantic communication under NLOS using dynamic shadows**, not generic NLOS reconstruction or standard gesture recognition. NLOS imaging work reconstructs hidden geometry or coarse presence but never uses NLOS shadows as a gesture-language channel; gesture recognition assumes LoS observation of visible hands, impossible in our “only wall shadow” setting. We now add a short subsection **“NLOS Gesture Decoding”** that explicitly states this intersection and positions RacoNet in that gap. “These methods” is replaced by “camera-driven NLOS imaging methods such as [Liu et al., 2024; Zhu et al., 2024; Czajkowski & Murray-Bruce, 2024]”.
>
> ---
>
> ### R2. Outer product expansion / tensor size / shape mismatch
>
> All computations use standard 2D feature maps, with complexity comparable to convolutions, so there is no tensor-size explosion. The reported “shape mismatch” is purely notational: some equations implicitly treated matrices as vectorized. We now add “vec(·)” where needed and provide a small notation table for dimensions. These changes affect only presentation, not the architecture or results.
>
> ---
>
> ### R3. Undefined symbols and directional filter design
>
> Several symbols were introduced too tersely. We now state that Eq. (2) uses directional filters at uniformly sampled angles, initialized as oriented gradient kernels and trained end-to-end, and we briefly justify the chosen number of directions. Line 242’s aggregation is clarified as a simple weighted sum of directional responses, not a tensor product. We explicitly define $S_m$ (number of multi-scale branches), $\zeta^{(m)}$ (per-channel spectral modulation vector), and $\kappa^{(m)}$ (small learned dynamic kernel predicted from local features), and move low-level implementation details to the appendix.
>
> ---
>
> ### R4. Dimensional symbols and shapes in ALT
>
> We unify dimensional notation: H,W for input; $H_s$,$W_s$ for ScLT feature sizes; $H′$,$W′$ for ALT input; $I_m$,$J_m$ for post-attention size; and $K_{(m,s)}$,$L_{(m,s)}$ for downsampled sizes at each scale. A short “notation summary” at the start of Sec. 3 lists all symbols. The ALT branch never increases resolution, so tensor sizes remain bounded.
>
> ---
>
> ### R5. Other undefined symbols
>
> Line 233: $\xi$ and $\eta$ denote horizontal and vertical coordinates on the relay surface in the 2D integral, distinct from x,y used elsewhere. Line 383: $\lambda_1$ and $\lambda_2$ are the weights for the geometric constraint and the main recognition loss; their values and ablations are already given, and we now define them explicitly on first use.
>
> ---
>
> ### R6. Dataset format / resolution / scale
>
> This is a presentation issue. We now give explicit statistics in Sec. 4.1 and a compact table. Briefly, S-Numbers and S-MNIST are derived from public sign-language image datasets passed through our RTE-based simulator, and S-MNISTm is obtained from a physical testbed with the same label space. For all three, we state image resolution, number of classes, sample counts and train/test splits, and clarify that images are resized to a common resolution at the model input.
>
> ---
>
> ### R7. Ablation is “too coarse”
>
> We disagree that the ablation is merely combinatorial. Beyond the main component ablation, the appendix includes module-level ablations (removing KA-ELNR or GIAO), loss-weight ablations over $\lambda_1:\lambda_2$, and illumination/noise robustness studies. We add a short “Ablation design rationale” paragraph linking each ablation to its design question.
>
> ---
>
> ### R8. Writing style / “physics-inspired” intuition
>
> The design is physics-driven, not post-hoc: RCLT corresponds to linear axial transfer vs. higher-order scattering, GIAO restores source geometry suppressed by diffuse reflections, and KA-ELNR implements subspace-wise nonlinear mappings in the spirit of Kolmogorov–Arnold. To improve readability, we add a brief **“Design overview & physical intuition”** subsection before Sec. 3, begin each method subsection with an intuitive paragraph, and shorten mixed physics/DL sentences.
>
> ---
>
> The issues raised are due to compact notation and writing rather than conceptual problems. We have clarified notation and datasets, tightened the link between ablations and design motivations, and reorganized the exposition to address these concerns.

---

### Official Review · Reviewer_tf9f · 2025-10-30

**Soundness:** 1
**Presentation:** 2
**Contribution:** 2
**Rating:** 2
**Confidence:** 4

**Summary:**

This paper studies NLOS vision by classifying hand gestures from indirect wall shadows. It proposes RacoNet, a physics-guided model combining light-transport constraints (RCLT) and a geometry-recovery module (GIAO), fused via KA-ELNR. Across three simulated sign-language datasets (plus one small measured setup), RacoNet beats general-purpose vision backbones, suggesting that physics-informed modeling helps on this synthetic shadow-classification task.

**Strengths:**

This paper propose a physics-aware deep architecture that explicitly models light transport and incorporates geometric reasoning, which is conceptually sound. This reflects a thoughtful attempt to bridge physical modeling with neural network design.

**Weaknesses:**

- The paper frames the work as NLOS decoding, but the core evaluation is closed-set classification of static hand gestures. NLOS problems typically prioritize reconstructing occluded geometry/appearance and then reasoning on top. Without any reconstruction, pose recovery, or even an interpretable intermediate, it’s hard to claim broader NLOS competence. A simple multi-task variant or silhouette/pose proxies would strengthen the case.

- Most baselines are general-purpose vision backbones that aren’t designed for NLOS. The paper should compare against passive NLOS pipelines that first reconstruct an image/silhouette/volume from wall observations and then classify. A two-step reconstruct-then-classify baseline might be competitive and would reveal whether RacoNet’s end-to-end approach is truly advantageous for NLOS.

- The main results rely on shadows simulated from sign-language datasets; the single measured set appears to be captured in a controlled testbed with fixed geometry. There is no evidence of performance in realistic conditions. Claims about cross-room communication remain speculative without demonstrations under varied, messy setups.

- Apart from limited lighting variation within the simulator, the paper does not stress test common NLOS failure modes: ambient flicker, dynamic illuminants, motion blur, defocus, rolling-shutter distortions, low-bit-depth quantization, mixed wall materials, or geometry shift. A principled robustness suite and cross-setup generalization study are needed.

**Questions:**

Please refer to weakness for more details.

---

### Official Review · Reviewer_Mv3c · 2025-10-31

**Soundness:** 3
**Presentation:** 3
**Contribution:** 3
**Rating:** 6
**Confidence:** 4

**Summary:**

This paper addresses the challenging problem of Non-Line-of-Sight (NLOS) gesture recognition by proposing a method to decode gestures solely from their highly distorted, dynamic shadows . The central contribution is a novel deep learning framework, named RacoNet, designed to interpret these subtle shadow variations. The key idea behind RacoNet is to decompose the complex inverse problem into modeling two distinct aspects: the physics of light propagation and the geometry of the occluded source . To this end, the framework incorporates: (1) a Radiance-Constrained Light-Transportation (RCLT) module, which further splits into two streams to separately model axial and scattered light paths ; (2) a Geometric Information Aliment Operation (GIAO) module to recover the latent geometry of the hand gesture ; and (3) a Kolmogorov-Arnold Enhanced Layerwise Nonlinear Reorganization (KA-ELNR) module to effectively fuse the physical and geometric features for the final classification . The authors validate their approach through experiments on both simulated (S-Numbers, S-MNIST) and physically captured (S-MNISTm) datasets, demonstrating that RacoNet significantly outperforms existing methods in decoding accuracy and robustness .

**Strengths:**

1. **Originality and Significance of the Problem:** I appreciate the problem posed in this paper—camera-based Non-Line-of-Sight (NLOS) gesture recognition. This task elegantly bridges the two interesting fields of NLOS perception and gesture recognition and has clear practical implications. To my knowledge, this is the first work to investigate this specific problem. I am pleased to see efforts aimed at extending the scope of passive NLOS tasks.

2. **Methodological Innovation:** I think the proposed Radiance-Constrained Light-Transportation (RCLT) module particularly novel. This module provides a well-reasoned decomposition of the NLOS projection problem into direct (axial) and scattered components. This is a clever way to introduce strong physical priors into the network architecture, and this physics-informed approach is a clear strength over generic vision models. While the novelty of the GIAO and KA-ELNR modules is less immediately apparent to me, the overall methodological framework remains innovative, primarily due to the RCLT design.

3. **Completeness of the Work:** The paper's appendix provides a solid mathematical formalism to illustrate the physical principles behind RCLT (e.g., Appendix A.1, A.2), which I greatly appreciate. Furthermore, the appendix includes an innovative extension to the decoding of continuous gesture sequences (via the text reassembly experiment in Appendix A.7), and also presents thorough robustness experiments (Appendix A.10, A.11). Overall, this makes the work feel well-rounded and comprehensive.

Collectively, these significant strengths in problem formulation and methodological design form the basis for my positive assessment of the paper.

**Weaknesses:**

However, the paper has significant weaknesses in other areas (primarily the two major concerns listed below), which make me to recommend a score of 4 (Borderline Accept). Should the authors be able to address these two primary concerns during the rebuttal phase, I would be happy to raise my score to 5-6.

1. **Generalization of the proposed network:**

   The experimental evaluation, while extensive in some aspects, critically lacks validation of the model's generalization capabilities. I think generalization is one of the most important aspect for any NLOS method due to the ill-posed nature of the NLOS problem. The current experiments are performed on datasets generated under specific, controlled conditions. It is unclear how the model would perform under slight variations, which are inevitable in real-world scenarios. Key unanswered questions include:

   * **Cross-Subject Generalization:** How does the model perform on gestures made by a person not seen during training?
   * **Scene Generalization:** How robust is the model to minor changes in the physical scene, such as a different camera position or light source angle?

   Without these experiments, it is difficult to assess whether the network has learned a generalizable physical model or has simply overfit to the specific training environment.

   Although the appendix gives robustness experiments on illumination (Appendix A.10) and noise (Appendix A.11), this is different from generalization experiments, because illumination and noise do not have an essential impact on the optical transport model, nor are they the main factors of variation in practical applications (different people and scenes).

   Therefore, I strongly suggest the authors to provide generalization results under different subjects and scene conditions in their rebuttal, as this will critically impact the evaluation of the proposed method.

2. **Clarity and Presentation:**

   The other primary weakness of the paper lies in its clarity, particularly in the introduction. The final paragraph of the introduction (Section 1), which is meant to outline the proposed method, is laden with highly technical and unexplained jargon (e.g., "high-order optical joint encoding," "missing latent space prior," "subspace decomposition nonlinear blocks"). This makes it exceedingly difficult, even for a reader familiar with the NLOS field, to grasp the core intuition and workflow of the proposed method without delving deep into the methodology section. This significantly hinders the accessibility and initial feeling of the paper.

   Furthermore, Figure 2, the main architecture diagram, is confusing as it fails to clearly illustrate the data flow between the GIAO module and the main RCLT-KA-ELNR pipeline.

   I suggest that the authors clearly state in their rebuttal how they plan to modify the Introduction and Figure 2 to improve readability.

3. **Minor Issues:**

   * The literature review in the Related Works section (Section 2.1) could be more comprehensive, as it appears to omit some classic and relevant works in passive NLOS imaging.
   * The use of the term "photonic propagation" is imprecise for a system using a conventional camera; "light propagation" or "optical propagation" would be more appropriate.

**Questions:**

See Weaknesses

---

### Official Review · Reviewer_XFgg · 2025-11-02

**Soundness:** 3
**Presentation:** 3
**Contribution:** 3
**Rating:** 6
**Confidence:** 2

**Summary:**

This paper explores the innovative direction of "achieving cross-room communication solely by decoding gesture shadows" and proposes a Radiation-constraint Network (RacoNet) to address the core challenges of dynamic shadow decoding. The key contributions are: designing the RCLT module to model axial and scattered light transmission via dual paths, accurately capturing light-space information; proposing the GIAO module to hierarchically recover the source-scene geometric structure, compensating for geometric loss caused by nonlinear transformations; constructing the KA-ELNR module to realize nonlinear fusion of light-space and geometric information. Experiments show that RacoNet achieves state-of-the-art performance on three datasets and exhibits robustness under illumination changes and noise interference, providing a new solution for non-line-of-sight communication, privacy-preserving gesture interaction and other scenarios.

**Strengths:**

1.Outstanding originality: For the first time, it systematically explores the feasibility of gesture shadows as a cross-room communication medium, proposing a complete framework of "physical modeling-geometric recovery-feature fusion". The design of three core modules specifically addresses the unique challenges of shadow decoding, with innovative research direction and technical route.
2.Excellent research quality: The experimental design is comprehensive, covering synthetic and measured datasets, verifying model performance through multiple metrics; ablation experiments detailedly validate the necessity of each module and key components; supplementary robustness experiments under illumination changes and noise interference enhance the credibility of conclusions.
3.Clear presentation: The paper has a coherent structure, progressing layer by layer from problem formulation, related work to method design and experimental verification; framework diagrams and ablation tables intuitively show the method flow and advantages, and technical details (such as frequency modulation of RCLT, hierarchical encoding of GIAO) are elaborated in detail for easy understanding.
4.Practical value: Focusing on practical needs such as non-line-of-sight communication and privacy protection, the results can be applied to disaster rescue, security surveillance, intelligent interaction and other scenarios, and provide new research ideas for shadow semantic decoding, NLOS imaging and other fields.

**Weaknesses:**

1.Limited scene adaptability: Experiments only verify the decoding effect of static gestures (digital and alphabet sign language) and do not involve dynamic continuous gesture sequences; the decoding performance under different distances and different diffusive surface materials is not evaluated, and the adaptability to real scenarios needs to be verified.
2.Suboptimal computational efficiency: RacoNet has a high parameter count (188.4M) and computational complexity (43.1G FLOPs), lacking efficiency advantages compared to lightweight models; no model lightweight schemes are explored, which is not conducive to deployment on edge devices.
3.Insufficient transparency of module design details: The nonlinear reorganization mechanism based on Kolmogorov-Arnold theory in the KA-ELNR module is described relatively abstractly, lacking clear mathematical derivation and intuitive visualization explanation; the fusion strategy of RCLT dual-path branches does not explain whether it dynamically adapts to different shadow scenarios.

**Questions:**

1.What is the performance of RacoNet in decoding dynamic continuous gesture sequences? Can supplementary experiments under different gesture speeds and sequence lengths be provided to illustrate its adaptability to dynamic scenarios?
2.Can supplementary decoding experiments under different propagation distances and different diffusive surface materials (such as walls, fabrics) be provided? What is the performance boundary of the model in extreme scenarios such as severely blurred shadows and multiple occlusions?
3.Compared with NLOS geometric recovery methods proposed after 2024 (such as models based on neural radiance fields), what are the advantages of RacoNet in decoding accuracy, robustness, and computational efficiency?
4.How sensitive is the model to shadow projection angle and light source intensity? Can relevant ablation experiments be supplemented to illustrate the impact of these factors on decoding performance?

---

### Meta-Review · Area_Chair_7Wkt · 2026-01-06

**Summary:**

**Summary**: This paper investigates the direction of achieving cross-room communication by decoding gesture shadows and proposes a Radiation-constraint Network (RacoNet) to address the core challenges of dynamic shadow decoding. The authors design three core modules within RacoNet: (1) the RCLT module, which models axial and scattered light transmission via dual paths to accurately capture light-space information; (2) the GIAO module, which hierarchically recovers the source-scene geometric structure; and (3) the KA-ELNR module, which enables nonlinear fusion of light-space and geometric information. Experimental results on three datasets demonstrate the effectiveness of the proposed method.

**Strengths**:
Reviewers praised the proposed method and tasks, noting that it elegantly bridges the fields of NLOS perception and gesture recognition, with clear practical implications. They also highlighted that several proposed modules, such as RCIL, are both interesting and novel. Furthermore, some reviewers acknowledged that the method has the potential to offer a new solution for non-line-of-sight communication, privacy-preserving gesture interaction, and related scenarios.

**Weaknesses**:
1.	Generalization capability and scene adaptability (Reviewers XFgg, Mv3c, tf9f, hbsw ): All reviewers raised significant concerns regarding the generalization capability of RacoNet, including its performance in diverse scenes, realistic conditions, decoding dynamic continuous gesture sequences, the model's performance boundary in extreme scenarios, and cross-subject generalization.
2.	Suboptimal computational efficiency (Reviewer XFgg)
3.	Clarity Issues (Reviewers XFgg, Mv3c, hbsw): The paper lacks clear mathematical derivations and intuitive visual explanations for the designed modules.  Reviewers also noted issues with the clarity of the introduction sections, and found the main figure confusing.
4.	Sensitivity to shadow projection angle and light source intensity (Reviewer XFgg)
5.	Insufficient comparisons (Reviewer tf9f): Comparative experiments are limited to general-purpose vision backbones.
6.	Insufficient related work section (Reviewers hbsw and Mv3c ): The related work section does not adequately clarify the connection between existing studies and this research.
7.	Insufficient ablation study (Reviewer hbsw)

**Decision**:
In the rebuttal, the authors only addressed some comments from Reviewer hbsw. However, they did not respond to the core concerns regarding limited scene adaptability and generalization capability, which were raised by almost all reviewers. Additional concerns—including clarity issues, sensitivity studies, the need for a more extensive ablation study, and suboptimal computational efficiency—were also not addressed. Therefore, I recommend rejection.

**Reviewer Concerns:**

In the rebuttal, the authors only addressed some comments from Reviewer hbsw. However, they did not respond to the core concerns regarding limited scene adaptability and generalization capability, which were raised by almost all reviewers. Additional concerns—including clarity issues, sensitivity studies, the need for a more extensive ablation study, and suboptimal computational efficiency—were also not addressed.

**Reviewer Scores:**

Reviewer XFgg (6 → slightly decreased or maintained, since no response), Reviewer Mv3c (6 → slightly decreased or maintained, since no response),  Reviewer tf9f (2 → maintained, since no response), Reviewer hbsw (4 → maintained, as only partial concerns were addressed)

---

### Decision · Program_Chairs · 2026-01-26

Reject